# NAB: Neural Adaptive Binning for Sparse-View CT reconstruction

**Wangduo Xie, Matthew B. Blaschko**
Center for Processing Speech and Images
Department of ESAT
KU Leuven
{wangduo.xie, matthew.blaschko}@esat.kuleuven.be

## Abstract

Computed Tomography (CT) plays a vital role in inspecting the internal structures of industrial objects. Furthermore, achieving high-quality CT reconstruction from sparse views is essential for reducing production costs. While classic implicit neural networks have shown promising results for sparse reconstruction, they are unable to leverage shape priors of objects. Motivated by the observation that numerous industrial objects exhibit rectangular structures, we propose a novel **N**eural **A**daptive **B**inning (**NAB**) method that effectively integrates rectangular priors into the reconstruction process. Specifically, our approach first maps coordinate space into a binned vector space. This mapping relies on an innovative binning mechanism based on differences between shifted hyperbolic tangent functions, with our extension enabling rotations around the input-plane normal vector. The resulting representations are then processed by a neural network to predict CT attenuation coefficients. This design enables end-to-end optimization of the encoding parameters—including position, size, steepness, and rotation—via gradient flow from the projection data, thus enhancing reconstruction accuracy. By adjusting the smoothness of the binning function, NAB can generalize to objects with more complex geometries. This research provides a new perspective on integrating shape priors into neural network-based reconstruction. Extensive experiments demonstrate that NAB achieves superior performance on two industrial datasets. It also maintains robust on medical datasets when the binning function is extended to more general expression. The code is available at https://github.com/Wangduo-Xie/NAB_CT_reconstruction.

## 1 Introduction

Computed Tomography (CT), as a high-precision non-destructive testing technology, plays an important role in industrial metrology, quality inspection, and medical diagnosis. Although the CT reconstruction accuracy typically improves with more scan views, acquiring additional views increases production costs in an industrial environment and radiation exposure in a medical scenario. Consequently, sparse-view CT reconstruction has become a key research problem. Moreover, since reconstruction is an ill-posed inverse problem, exploring how to perform CT reconstruction at fewer views is also crucial for uncovering deeper mathematical structures.

With the advancement of deep learning, supervised learning has been used to construct pairs of sparse views and dense views to learn their correspondence. However, due to the data offset between the training and testing datasets, such methods often suffer from limited generalization ability. To avoid this problem, Implicit Neural Representations (INRs) have been used for self-supervised CT reconstruction by encoding spatial coordinates into Fourier features, which are then mapped to the attenuation coefficient using a neural network, relying solely on the object's own projection data.

Despite their success in CT reconstruction, INRs struggle to capture the shape prior presented in industrial CT scans and are fundamentally restricted by its limited harmonic representation. In particular, industrial CT often involves man-made objects with predominantly rectangular structures—such as bricks, brackets, or metal plates. This structure serves as a powerful prior, yet most existing INR

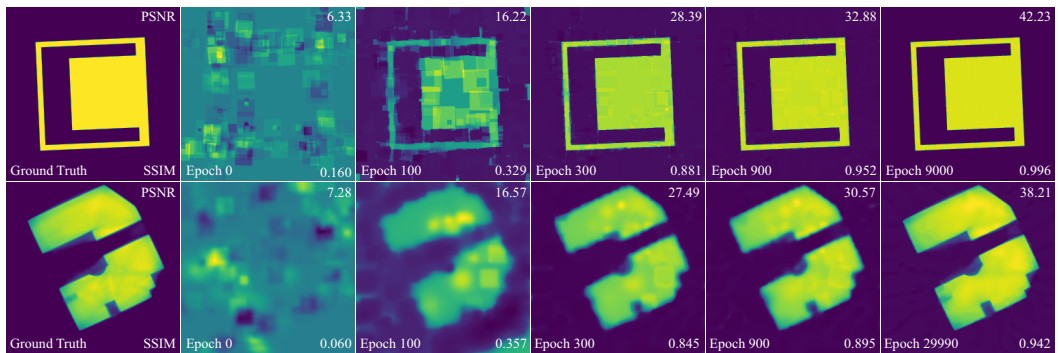

Figure 1: Evolution of adaptive binning during training using our self-supervised method on two different samples (16 views). The reconstructed CT images gradually approach the ground truth.

variants fail to leverage it effectively. To address this, we propose a method that explicitly utilizes this shape prior, enabling more accurate reconstruction. Furthermore, by relaxing the assumption of rectangularity, our algorithm can generalize to broader geometries beyond rectangular forms.

Specifically, instead of encoding each coordinate to the random Fourier features, we propose a novel scheme named neural adaptive binning (NAB). In this approach, each dimension of the feature vector corresponds to a specific binning operation over the coordinate plane. To construct the binning operation, we first compute the height difference of the shifted hyperbolic tangent function along one axis. We then apply a similar procedure in the orthogonal direction. These difference functions, orientated along different directions, are then combined via Hadamard product to form a localized bin. However, since rectangular regions in the real world usually appear in various orientations rather than being strictly axis-aligned, we introduce an affine transformation in the input coordinate space before calculating the difference functions. This allows the resulting bin to be rotated in a differentiable manner, enabling our model to adaptively capture orientation variations.

Due to the differentiability of the hyperbolic tangent function, our binning scheme supports not only rigid body transformation but also scaling transformation. Once the encoding is completed, a neural network maps the binning representation to the CT attenuation coefficient for reconstruction, as shown in Figure 1. To further enhance flexibility, we introduce a multi-scale steepness mechanism that generalizes the rectangular binning to smoother, bulge-like binning structures. The extension enables the framework to reconstruct objects with both rectangular and curved geometries. Moreover, we analytically derive the upper bound of the distance between our NAB representation and a set of binary vectors, and prove that as the steepness approaches infinity, the distance converges to zero. The convergence proof builds a theoretical connection between our NAB and the random binning scheme in kernel methods. In summary, our contributions are as follows:

❑ (1) Our work proposes a novel encoding method, Neural Adaptive Binning (NAB), which explicitly models rectangular priors commonly observed in industrial CT scenarios and offers a new perspective on shape-aware representation learning.

❑ (2) We designed a differentiable way that adaptively adjusts the size, position, and rotation of each bin, enabling end-to-end optimization of bin parameters alongside the neural network. Furthermore, we propose a multi-scale steepness strategy to accommodate both rectangular and curved structures. Through function approximation analysis, we build a bridge between NAB and random hard bins.

❑ (3) We conduct comprehensive ablation studies to analyze the impact of different binning components in NAB, and perform extensive comparative experiments against leading methods, demonstrating the superior performance of our approach.

## 2 RELATED WORK

**Classic CT reconstruction** Classical CT reconstruction techniques can be broadly divided into analytical and iterative methods. Among them, FBP (Herman, 2009) is the representative of analytical

methods. Iterative methods are mainly composed of SIRT (Gregor & Benson, 2008), SART (Andersen & Kak, 1984), and NAG_LS (Nesterov, 2013; Neubauer, 2017). In addition, there are also iterative methods such as Wang & Jiang (2004); Sidky & Pan (2008). Analytical methods are faster but prone to artifacts, while iterative methods are slower but have fewer artifacts.

**Self-supervised CT reconstruction** Due to the generalization problem of supervised methods, self-supervised neural networks have broad prospects in the field of CT reconstruction. Their representatives are deep image prior (DIP) (Ulyanov et al., 2018; Baguer et al., 2020) and implicit neural representation (INR) (Tancik et al., 2020; Sitzmann et al., 2020). The classical INR encodes the coordinates as trigonometric functions, which are then combined by a neural network to reconstruct the object. This type of method has achieved good performance in rendering (Mildenhall et al., 2021) and reconstruction (Niemeyer et al., 2020) in natural scenes. And it has gradually become popular in the field of CT reconstruction. For CT reconstruction tasks, most methods directly use INRs without external priors (Sun et al., 2021; Zang et al., 2021; Rückert et al., 2022; Zha et al., 2022; Wu et al., 2023b; Saragadam et al., 2023; Wu et al., 2023a; Cai et al., 2024). There has been recent works combining external CT data or material priors into INR (Shen et al., 2022; Gu et al., 2023; Shi et al., 2024; Liu et al., 2024; Liu & Bai, 2024; Xie et al., 2025; Tian et al., 2025). However, these methods overlook the role of coordinate encoding in unsupervised prior acquisition. In addition, when changing the representation from implicit to explicit, some methods use 3D Gaussian (Kerbl et al., 2023) for CT reconstruction (Zha et al., 2024; Li et al., 2025; Yuluo et al., 2025) in explicit representation. However, 3D Gaussian-based methods lack strong fitting ability compared to neural network, resulting in a loss of accuracy. In addition, restricted to the Gaussian form, they cannot model rectangular patterns explicitly.

## 3 MOTIVATION

INRs with Random Fourier Coding (RFC) often generate distinctive wave-like artifacts during reconstruction. The issue arises from two main factors. First, RFC is based on random sampling, which prevents the INR from using an object's shape prior. Second, the INR is limited to certain combinations of harmonics. An INR with RFC is limited to representing functions of the form: (Yüce et al., 2022)

$$\sum_{\omega \in \mathcal{H}(\Omega)} c_\omega \sin((\omega, r) + \phi_\omega) \tag{1}$$

Among these, $\mathcal{H}(\Omega)$ is limited by the RFC frequency matrix $\Omega := [\Omega_0, ..., \Omega_{T-1}]^\top$, which is sampled once and fixed before INR training. The restriction takes the following form:

$$\mathcal{H}(\Omega) \subseteq \left\{ \sum_{t=0}^{T-1} s_t \Omega_t \ \middle| \ s_t \in \mathbb{Z} \wedge \sum_{t=0}^{T-1} |s_t| \leq K^{L-1} \right\} \tag{2}$$

Intuitively, training an INR can be regarded as making transitions among the functions that can be represented by Equation (1). Consequently, when the object goes beyond the range specified

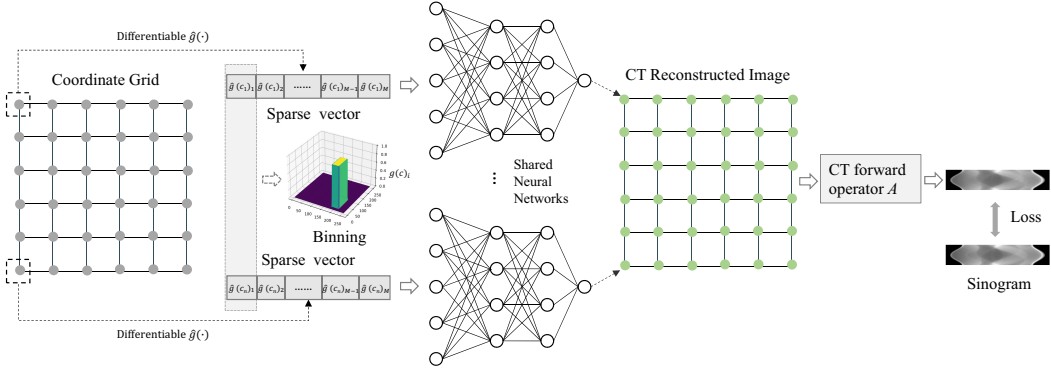

Figure 2: Our method's overall framework.

by Equation (1), artifacts are likely to emerge. Moreover, according to the Gibbs phenomenon, an overshoot in height inevitably occurs near a jump discontinuity point, even as the number of trigonometric functions in the approximation approaches infinity. Therefore, to better handle rectangular boundaries in industrial objects, and incorporate the shape prior, we abandon the RFC and instead propose a new method based on adaptive binning coding.

# 4 METHODOLOGY

## 4.1 PARALLEL BINNING DESIGN

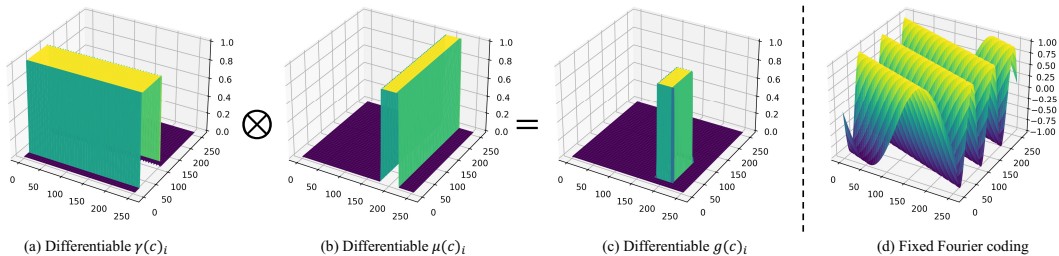

(a) Differentiable $\gamma(c)_i$     (b) Differentiable $\mu(c)_i$     (c) Differentiable $g(c)_i$     (d) Fixed Fourier coding

Figure 3: Differentiable Binning Features $g(c)_i$ and its generation way in our method (subgraph (a), (b), (c)) and Fixed Fourier Feature in classic INR (subgraph (d)). The $\otimes$ represents Hadamard product, which will act on two $256 \times 256$ matrices.

For any coordinate vector $c \in \mathbb{R}^N$ in an $N$-dimensional coordinate system, our objective is to construct a differentiable $f_E$ that maps $c$ to vector representation $v_c$, with the explicit purpose of leveraging the industrial object's rectangular prior:

$$v_c := f_E(c) \tag{3}$$

However, there is no natural differentiable function that can directly express the prior. To circumvent this challenge, we design square waves along $N$ orthogonal directions. Without loss of generality, we illustrate our method with the case of $N = 2$. The square waves $\gamma(c)_i$ along the $x$-axis are constructed as the difference between two shifted hyperbolic tangent functions on the same axis:

$$\gamma(c)_i := \gamma(c)_{left_i} - \gamma(c)_{right_i} \tag{4}$$

where:

$$\gamma(c)_{left_i} := \frac{1}{2}tanh(k_i(x_c - u_i + \frac{1}{2}h_i)) \tag{5}$$

$$\gamma(c)_{right_i} := \frac{1}{2}tanh(k_i(x_c - u_i - \frac{1}{2}h_i)) \tag{6}$$

where $x_c$ denotes $x$-axis coordinate of $c$, and $k_i$ controls the steepness of the $tanh$ function. The parameter $u_i$ specifies the center of the first parallel shift of the $tanh$ function, while $\frac{1}{2}h_i$ and $-\frac{1}{2}h_i$ correspond to the distance of the second and third parallel displacement, respectively. An instance of square waves, $\gamma(c)_i$, is depicted in subgraph (a) of Figure 3. By duality, we construct another square wave $\mu(c)_i$ along the orthogonal direction (the $y$-axis):

$$\mu(c)_i := \mu(c)_{left_i} - \mu(c)_{right_i} \tag{7}$$

where:

$$\mu(c)_{left_i} := \frac{1}{2}tanh(k_i(y_c - v_i + \frac{1}{2}w_i)) \tag{8}$$

$$\mu(c)_{right_i} := \frac{1}{2}tanh(k_i(y_c - v_i - \frac{1}{2}w_i)) \tag{9}$$

where $y_c$ represent the $c$'s $y$-axis coordinate. Similar to the definition of $x$-axis direction, $v_i$, $\frac{1}{2}w_i$ and $-\frac{1}{2}w_i$ denotes the center and displacement along the $y$-axis direction.

To obtain a bin of length $h_i$ and width $w_i$, centered at $(u_i, v_i)$, we take the product of the square waves $\mu(c)_i$ and $\gamma(c)_i$ from the two orthogonal directions:

$$g(c)_i := \mu(c)_i \times \gamma(c)_i \qquad (10)$$

The visualization of $g(c)_i$ is shown in subgraph (c) of Figure 3. Until now, a rectangular shape can be approximated by combining multiple bins with adaptive translation and scaling; however, this approach yields suboptimal performance when the object contains oblique rectangular regions. Therefore, we propose a novel rotation embedding strategy in Section 4.2.

## 4.2 ROTATION TRANSFORMATION

To relax the restriction that bins must be axis-aligned, we generalize the bin's representation $g(c)_i$ by incorporating rotations with arbitrary angle $\theta_i$. The rotated representation is thus defined as $\hat{g}(c)_i$:

$$\hat{g}(c)_i := (\hat{\gamma}(c)_{left_i} - \hat{\gamma}(c)_{right_i}) \times (\hat{\mu}(c)_{left_i} - \hat{\mu}(c)_{right_i}) \qquad (11)$$

where $\hat{\gamma}(c)_{left_i}$ and $\hat{\gamma}(c)_{right_i}$ denote the results of rotating $\gamma(c)_{left_i}$ and $\gamma(c)_{right_i}$ along the normal direction of the input coordinate plane, with the rotation center located at $(u_i, v_i)$:

$$\hat{\gamma}(c)_{left_i} := \frac{1}{2}[tanh(k_i([cos(\theta_i), -sin(\theta_i)][x_c - u_i, y_c - v_i]^\top + \frac{1}{2}h_i)) \qquad (12)$$

$$\hat{\gamma}(c)_{right_i} := \frac{1}{2}[tanh(k_i([cos(\theta_i), -sin(\theta_i)][x_c - u_i, y_c - v_i]^\top - \frac{1}{2}h_i)) \qquad (13)$$

Figure 8 in Appendix illustrates the rotation of $\hat{\gamma}(c)_{right_i}$ at different angles. Similarly, the results of rotating $\mu(c)_{left_i}$ and $\mu(c)_{right_i}$ around $(u_i, v_i)$ in the same direction are formulated as follows:

$$\hat{\mu}(c)_{left_i} := \frac{1}{2}[tanh(k_i([sin(\theta_i), cos(\theta_i)][x_c - u_i, y_c - v_i]^\top + \frac{1}{2}w_i)) \qquad (14)$$

$$\hat{\mu}(c)_{right_i} := \frac{1}{2}[tanh(k_i([sin(\theta_i), cos(\theta_i)][x_c - u_i, y_c - v_i]^\top - \frac{1}{2}w_i)) \qquad (15)$$

By rotating the different components $\mu(\cdot)$ and $\gamma(\cdot)$ to $\hat{\mu}(\cdot)$ and $\hat{\gamma}(\cdot)$, respectively, we effectively rotate $g(c)_i$ around $(u_i, v_i)$ by angle $\theta_i$, yielding $\hat{g}(c)_i$ (see Equation (11)). Differentiable angle $\theta_i$ can be learned during self-supervised CT reconstruction. Finally, by introducing the magnification (height) factor $\lambda_i$ for each bin, the $f_E(c)$ in Equation (3) is designed as:

$$f_E(c) := [\lambda_1 \hat{g}(c)_1, \lambda_2 \hat{g}(c)_2, ..., \lambda_M \hat{g}(c)_M]^\top. \qquad (16)$$

## 4.3 LIMITING APPROXIMATION OF BINNING

Through the above method, every coordinate $c$ is mapped to a vector $f_E(c)$. When $\lambda_i = 1.0$, $f_E(c)$ lies close to the binary vector set $S := \{(h_1, h_2, ..., h_M)^\top | h_i \in \{0, 1\}\}$. Interestingly, the distance converges to 0 as all $k_i$ increase. Specifically, for every $c$, the distance between $f_E(c)$ and $S$ is controlled by a function $f_{dis} : \mathbb{R}^{6M} \to \mathbb{R}^+$ whose variables are $\{k_i, u_i, v_i, w_i, h_i, \theta_i | i = 1, 2, ...M\}$:

$$\min_{z \in S} ||f_E(c) - z||_1 \leq f_{dis}(\{k_i, u_i, v_i, w_i, h_i, \theta_i | i = 1, 2, ...M\}) \qquad (17)$$

For any given parameters set $\{u_i, v_i, w_i, h_i, \theta_i | i = 1, 2, ...M\}$, $f_{dis}$ has the properties:

$$\lim_{k_1, ..., k_M \to +\infty} f_{dis}(\{k_i, u_i, v_i, w_i, h_i, \theta_i | i = 1, 2, ...M\}) = 0. \qquad (18)$$

By the squeeze theorem, we know that for every $c$, the following holds:

$$\lim_{k_1, ..., k_M \to +\infty} \min_{z \in S} ||f_E(c) - z||_1 = 0 \qquad (19)$$

When Equation (19) is enforced, it guarantees that the bin contains the distinct boundary. As a result, $f_E(c)$ can represent ideal rectangular bins. This can also be viewed as a type of hard binning feature (Rahimi & Recht, 2007), albeit in a non-random manner.

### 4.4 Link to Neural Networks

For every $c \in \mathbb{R}^N$, we employ a neural network $f_{net} : \mathbb{R}^M \to \mathbb{R}^1$ to map the adaptive binning features $f_E(c) \in \mathbb{R}^M$ to the CT attenuation coefficient at coordinate $c$ within the object:

$$X_c := f_{net}(f_E(c)) \tag{20}$$

The overall framework of our algorithm is shown in Figure 2. The loss function during training is calculated on the projection domain:

$$Loss := \|A(X) - Y\|_2^2 \tag{21}$$

Where $X \in \mathbb{R}^{H \times W}$ represents the object to be optimized by spreading $X_c \in \mathbb{R}^1$ over the entire coordinate grid according to coordinate $c$. $A : \mathbb{R}^{H \times W} \to \mathbb{R}^{U \times V}$ represents the forward operator, which is controlled by parameters of imaging geometry and machine configuration with $V$ detector pixels and $U$ rotations. $Y \in \mathbb{R}^{U \times V}$ represents the collected sinogram data.

When minimizing Equation (21), the weights and biases of $f_{net}$ and differentiable parameter set $\{\lambda_i, k_i, \theta_i, u_i, v_i, h_i, w_i | i = 1, 2, ..., M\}$ of $f_E$ will be updated at the same time.

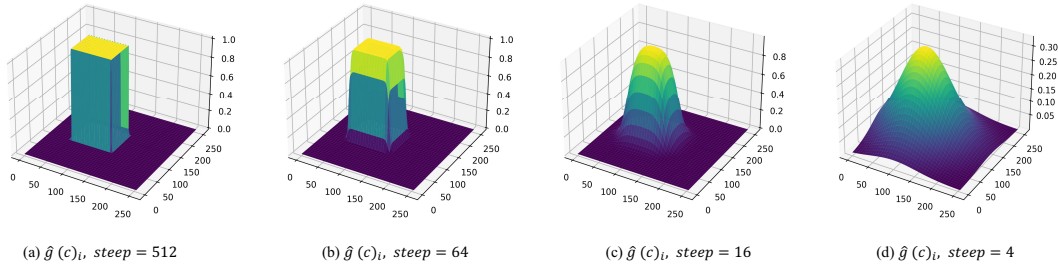

(a) $\hat{g}(c)_i$, $steep = 512$     (b) $\hat{g}(c)_i$, $steep = 64$     (c) $\hat{g}(c)_i$, $steep = 16$     (d) $\hat{g}(c)_i$, $steep = 4$

Figure 4: Differentiable Binning Features $\hat{g}(c)_i$'s multi-scale variations with changing steepness $k_i$.

### 4.5 Shape Generalization

Our method is mainly designed for rectangular regions commonly encountered in industrial CT. However, the proposed binning function, as an approximate bump function, is also capable of representing more complex geometry. For objects comprising both rectangular and non-rectangular geometry, the bin is configured using a multi-scale steepness approach that incorporates finer steepness levels. Specifically, each $k_i$ is taken from the set $\{p_1, ..., p_q\}$, which includes both large and small steepness, where $q > 1$ denotes the number of scales. As shown in Figure 4, different steepness values lead to different smooth variants. These variants provide rich bases for reconstructing objects. We further validate the effectiveness of the multi-scale mechanism in Section 5, which involves a dataset containing regions with non-zero curvature.

## 5 Experiments

### 5.1 Dataset and Implementation Details

**Dataset** We extracted slice from a calcium carbonate hollow cube (Schoonhoven et al., 2024) and rotated it at 19 different angles to create 19 different phantoms. From these, we generated CT projections from 16, 14, and 12 views for each phantom, respectively, which we refer to as the *CaCO₃ dataset*. In addition, we sample 10 slices with diverse structures from the Zeiss data (Klacansky, 2017) and generate CT projections with the same view configurations as the *CaCO₃ dataset*, forming the *Workpieces dataset*. All projections were acquired under a parallel-beam geometry. In addition to industrial datasets, we also conducted experiments on medical datasets, the results are shown in Appendix Section A.8.

**Implementation Details** Our algorithm is based on the PyTorch framework, where the optimizer is the Adam algorithm with $(\beta_1, \beta_2) = (0.9, 0.999)$. The initialization strategy and learning rate

setting are described in Appendix Section A.4 and Section A.5. For the $CaCO_3$ dataset, the multi-scale steepness values are set to $\{600, 800\}$ and for Workpieces dataset, they are set to $\{25, 50, 75\}$. We use ReLU as the activation function in our method and fix the length of the adaptive binning feature to 456. A four-layer neural network is used in our approaches. Our method stops training at epoch 29,990, and we report results at both epoch 19,990 and 29,990. The differentiability of the operator $A$ in Equation (21) is implemented using Tomosipo (Hendriksen et al., 2021).

## 5.2 PERFORMANCE COMPARISON

**Comparison methods** Since most competitive reconstruction methods are based on INRs, we primarily compare our method with them. Specifically, we compare $INR_f$, an INR based on random Fourier feature encoding (Tancik et al., 2020), implemented with a four-layer fully connected network (FCN) whose width, depth, and activation function are identical to those of the FCN in our method. In addition, the length of the Fourier features in the $INR_f$ is set to 456, the same as our bin encoding length. However, the number of parameters in $INR_f$ is $3 \times 10^3$ fewer than in our method. To control the parameters' effect on performance, we introduce another baseline: $INR_{l_1}$, which uses 260 neurons in each hidden layer—resulting in $3 \times 10^3$ more parameters than our method.

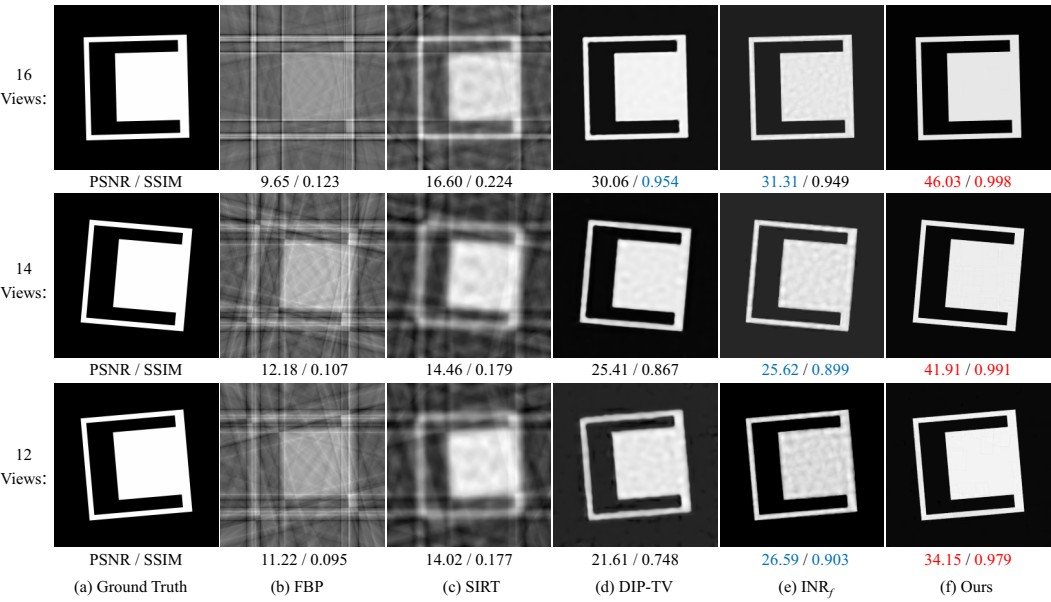

| | | | | | |
|---|---|---|---|---|---|
| 16 Views: | | | | | |
| PSNR / SSIM | 9.65 / 0.123 | 16.60 / 0.224 | 30.06 / 0.954 | 31.31 / 0.949 | 46.03 / 0.998 |
| 14 Views: | | | | | |
| PSNR / SSIM | 12.18 / 0.107 | 14.46 / 0.179 | 25.41 / 0.867 | 25.62 / 0.899 | 41.91 / 0.991 |
| 12 Views: | | | | | |
| PSNR / SSIM | 11.22 / 0.095 | 14.02 / 0.177 | 21.61 / 0.748 | 26.59 / 0.903 | 34.15 / 0.979 |
| (a) Ground Truth | (b) FBP | (c) SIRT | (d) DIP-TV | (e) $INR_f$ | (f) Ours |

Figure 5: Reconstruction at 12, 14, 16 views on the $CaCO_3$ dataset. The best numerical result is marked in red, and the second-best result is marked in blue. **Zooming** in shows that the comparison methods exhibit edge blurring and distortions, while our method maintains sharp boundaries.

We also compared our method with $Siren_1$, which replaces the activation function in the neural network of $INR_f$ with a sine function. Furthermore, we compared our method with $Siren_2$, which, consistent with (Sitzmann et al., 2020), does not use positional encoding like Random Fourier Coding. $Siren_2$ consists of five layers: 456 neurons in the first hidden layer and 260 neurons in the remaining hidden layers, all using sine activation functions. In addition, we include comparisons with leading unsupervised methods, including Instant-NGP (Müller et al., 2022; Zha et al., 2022), DIP (Ulyanov et al., 2018; Baguer et al., 2020), DIP-TV (Liu et al., 2019) which adds an isotropic TV loss to DIP in the image domain, and Wire (Saragadam et al., 2023), as well as classical methods FBP, SIRT, and NAG_LS. For deep-learning based comparative methods that rely on implicit representations, we report results at 29,990 epochs, which is equal to our method's training epochs. However, for DIP and DIP-TV, we adopt an earlier stopping point than 29,990 epochs, since these methods are more prone to overfitting.

Table 1: Numerical results of reconstruction on the CaCO$_3$ dataset. "Trainable Params" refers to the total number of trainable parameters. The best result is in **bold** and the second-best is underlined.

| Methods | 16-view | | 14-view | | 12-view | | Trainable |
|---|---|---|---|---|---|---|---|
| | PSNR↑ | SSIM↑ | PSNR↑ | SSIM↑ | PSNR↑ | SSIM↑ | Params↓ |
| FBP | 11.74 | 0.125 | 10.68 | 0.109 | 9.44 | 0.095 | - |
| SIRT | 15.64 | 0.201 | 15.14 | 0.195 | 14.75 | 0.195 | - |
| NAG_LS | 15.99 | 0.204 | 15.48 | 0.197 | 15.11 | 0.196 | - |
| DIP | 22.42 | 0.673 | 21.69 | 0.651 | 20.89 | 0.563 | $1.90 \times 10^6$ |
| DIP-TV | 24.05 | 0.783 | 21.98 | 0.703 | 22.40 | 0.688 | $1.90 \times 10^6$ |
| Instant-NGP | 30.81 | 0.953 | 30.03 | 0.946 | 27.23 | **0.911** | $2.96 \times 10^6$ |
| Wire | 23.80 | 0.785 | 23.35 | 0.752 | 21.96 | 0.721 | $2.65 \times 10^5$ |
| Siren$_1$ | 19.82 | 0.227 | 18.34 | 0.193 | 16.98 | 0.164 | $\mathbf{2.49 \times 10^5}$ |
| Siren$_2$ | 19.64 | 0.210 | 18.57 | 0.186 | 17.67 | 0.175 | $2.56 \times 10^5$ |
| INR$_f$ | 29.01 | 0.934 | 27.10 | 0.891 | 25.08 | 0.854 | $\mathbf{2.49 \times 10^5}$ |
| INR$_{l_1}$ | 27.94 | 0.901 | 25.55 | 0.831 | 22.82 | 0.750 | $2.55 \times 10^5$ |
| INR$_{l_2}$ | 38.89 | 0.983 | 37.83 | 0.970 | 30.36 | 0.901 | $1.25 \times 10^6$ |
| Ours (Iter = 19990) | 41.52 | 0.991 | 36.09 | 0.945 | 30.51 | 0.853 | $\underline{2.52 \times 10^5}$ |
| Ours (Iter = 29990) | **43.61** | **0.996** | **40.07** | **0.977** | **34.72** | 0.888 | $\underline{2.52 \times 10^5}$ |

Since the compared methods exhibit limited performance on the CaCO$_3$ dataset, we additionally evaluate INR$_{l_2}$, an INR based on a seven-layer FCN in which each hidden layer has 456 neurons. **Notably, INR$_{l_2}$ has four times more parameters than our method**.

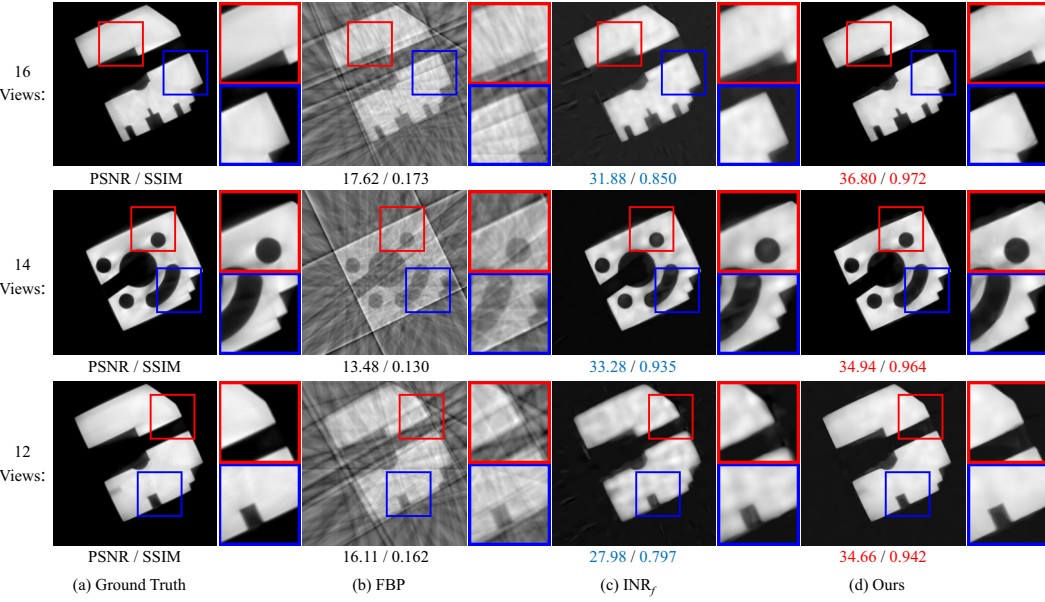

Figure 6: Reconstruction at 12, 14, 16 sparse views on the Workpieces dataset.

**Quantitative evaluation** For every method, Table 1 reports the average PSNR/SSIM (Appendix Section A.6) on the CaCO$_3$ dataset. With the same FCN architecture, replacing random Fourier encoding with our NAB boosts INR$_f$ by 9.64 dB, 12.97 dB and 14.60 dB under 12, 14 and 16 view settings, respectively. For INR$_{l_1}$, although it has more parameters than our method, its performance remains inferior to our method. When INR$_{l_1}$ is expanded to INR$_{l_2}$, which has nearly five times as many parameters as our model, our method still outperforms it by 4.36 dB, 2.24 dB, and 4.72 dB under 12, 14 and 16 views, respectively. Figure 7 shows that, under both the 14 views and 16 views settings, **our method is better than INR$_f$ and INR$_{l_2}$ in almost every epoch**.

For the $CaCO_3$ dataset, Instant-NGP achieved the highest performance in the 12 view SSIM metric, however, it exhibited lower PSNR values than our method across all views. The substantial number of trainable parameters and the limited interpretability remain inherent limitations of the Instant-NGP. In comparison, our method has fewer parameters and better interpretability.

For the Workpieces dataset, Table 2 summarizes the performance of different methods. It shows that our method exceeds $INR_f$ by 5.39 dB, 1.20 dB and 2.92 dB under 12, 14, and 16 views, respectively. For $INR_{l_1}$, our method outperforms it by an average of 2.82 dB across different settings. The performance improvement on this dataset is smaller than that on the $CaCO_3$ dataset, mainly because the Workpieces dataset contains not only rectangular, but also many curved shapes like arcs and circles. Even so, our method still achieves the best reconstruction performance with a small number of parameters. Meanwhile, we observe a performance reversal at the 14 view and 16 view settings for DIP, Instant-NGP, $INR_f$, and $INR_{l_1}$, where the 14 view results are superior to 16 view configuration. We attribute this phenomenon to the consistent orientation of phantoms in the Workpieces dataset, which introduces a coupling effect between the reconstruction algorithms and their orientation sensitivity. In addition, we find that our method achieves competitive performance while requiring 10k fewer training epochs, showing that it works well with fewer iterations.

Table 2: Numerical results of reconstruction on the Workpieces dataset. "Trainable Params" refers to the total number of trainable parameters. The best result is in **bold** and the second-best is underlined.

| Methods | 16-view | | 14-view | | 12-view | | Trainable |
|---|---|---|---|---|---|---|---|
| | PSNR↑ | SSIM↑ | PSNR↑ | SSIM↑ | PSNR↑ | SSIM↑ | Params↓ |
| FBP | 18.12 | 0.198 | 15.48 | 0.165 | 16.61 | 0.165 | - |
| SIRT | 22.91 | 0.385 | 22.68 | 0.382 | 21.04 | 0.347 | - |
| NAG_LS | 23.29 | 0.381 | 23.15 | 0.373 | 21.37 | 0.348 | - |
| DIP | 28.28 | 0.713 | 29.41 | 0.704 | 25.76 | 0.625 | $1.90 \times 10^6$ |
| DIP-TV | 33.64 | 0.918 | 29.89 | 0.777 | 28.20 | 0.786 | $1.90 \times 10^6$ |
| Instant-NGP | 31.17 | 0.904 | 31.95 | 0.914 | 28.38 | 0.832 | $2.96 \times 10^6$ |
| Wire | 28.77 | 0.834 | 28.14 | 0.796 | 27.29 | 0.794 | $2.65 \times 10^5$ |
| $Siren_1$ | 31.56 | 0.614 | 30.03 | 0.541 | 29.11 | 0.564 | $\mathbf{2.49 \times 10^5}$ |
| $Siren_2$ | 32.70 | 0.700 | 31.82 | 0.639 | 28.89 | 0.590 | $2.56 \times 10^5$ |
| $INR_f$ | 33.34 | 0.911 | 34.03 | 0.912 | 27.37 | 0.787 | $\mathbf{2.49 \times 10^5}$ |
| $INR_{l_1}$ | 33.89 | 0.929 | 34.11 | 0.926 | 27.80 | 0.828 | $2.55 \times 10^5$ |
| Ours (Iter = 19990) | 35.70 | 0.926 | 34.98 | 0.927 | 31.85 | 0.896 | $2.52 \times 10^5$ |
| Ours (Iter = 29990) | **36.26** | **0.938** | **35.23** | **0.941** | **32.76** | **0.909** | $2.52 \times 10^5$ |

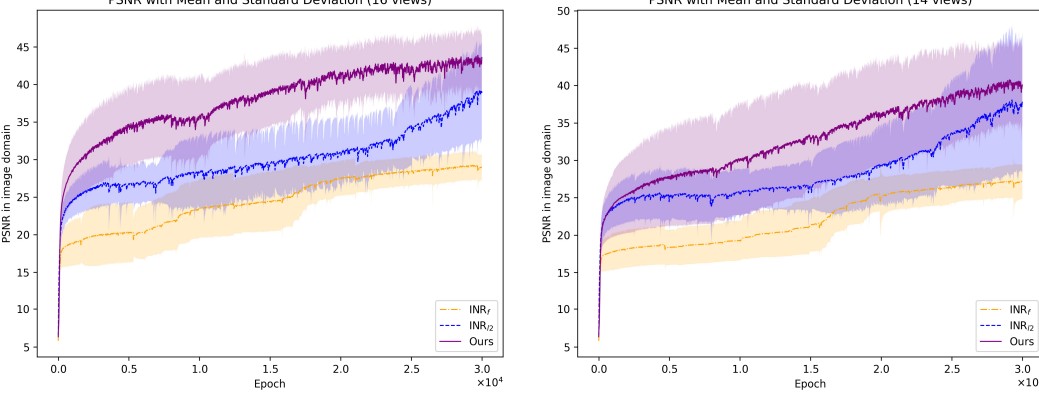

Figure 7: Avg. PSNR ± std. per epoch on the $CaCO_3$ dataset(14 views and 16 views). Our method outperforms $INR_f$ and $INR_{l_2}$ in almost every epoch, while using only 20% of $INR_{l_2}$'s parameters.

**Qualitative evaluation** The qualitative evaluation of reconstruction is shown in Figure 5 and Figure 6. Since this paper primarily compares the proposed encoding method and Fourier encoding, the visualization includes two representative instances ("$INR_f$" and "Ours"). For both instances, the neural network is kept unchanged, and only the encoding method is switched from Fourier encoding to NAB. For the $CaCO_3$ dataset, our method reconstructs sharp edges with minimal artifacts, while the comparison methods all exhibit blurred edges and distortion. The triangular ripple artifacts of the $INR_f$ illustrate its difficulty in reconstructing rectangular areas. The phenomenon reflects the limitations of random Fourier coding compared to our NAB. In addition, we have included a visualization comparing $INR_{l_2}$ and our method in the Section A.7. For the Workpieces dataset, we can see that our method not only reconstructs straight edges, but also reconstructs edges with non-zero curvature. Furthermore, our method produces fewer artifacts in non-edge areas on this dataset.

## 5.3 ABLATION STUDY

To verify the effectiveness of the different differentiable parts in our method, we conducted an ablation experiment as shown in Table 3. Firstly, the results show that the bin's center location $\{u_i, v_i\}$ and bin's side length $\{h_i, w_i\}$ are the most important parts. If the algorithm makes them fixed, it will result in a 10+ dB performance degradation. Secondly, the bin's ability to rotate is the second most important part. Once the rotation ability is disabled, the average PSNR on the entire dataset will decrease by 6.11 dB. Thirdly, the loss of the automatic adjustment of steepness will also affect the final performance, resulting in a 0.7 dB performance degradation. Finally, the trainability of bin height has only a 0.24 dB impact on performance, suggesting that it is nearly compensated by the subsequent neural network.

Table 3: Ablation study on the $CaCO_3$ dataset (16 views). "w/o $X$" means that the component "$X$" is kept frozen (not updated) during reconstruction, with all other components remaining trainable.

| Methods | PSNR↑ | SSIM↑ |
|---|---|---|
| w/o Center Location | 29.82 | 0.901 |
| w/o Side Length | 30.82 | 0.787 |
| w/o Bin Rotation | 37.50 | 0.982 |
| w/o Bin Height | 43.37 | 0.996 |
| w/o Steepness | 42.91 | 0.994 |
| Full setting (Ours) | **43.61** | **0.996** |

## 6 CONCLUSION

This paper proposes a novel CT reconstruction method based on neural adaptive binning (NAB). We design the mathematical model of binning as the difference between two shifted hyperbolic tangent functions. By embedding rotational transformations into the bin's expression and making the bin's parameters differentiable, we generalize the vanilla bin into a flexible binning encoding that can adaptively scale, rotate, and shift to appropriate variations. This encoding can not only capture rectangular patterns but also adjust to shapes with non-zero curvature. After the binning representation, the encoding results are further mapped to CT attenuation coefficients via a subsequent neural network. Under the supervision of projection data, our algorithm can calculate the final CT reconstruction in a self-supervised manner. Furthermore, we prove that the mathematical limits of this encoding result in strict coordinate partitioning and have a close relationship with random hard bins. Extensive experiments on two industrial datasets demonstrate that our method effectively leverages shape priors and achieves superior reconstruction performance. In addition, it also performs robustly on medical datasets when the binning function is generalized to smoother formulation.

## ACKNOWLEDGMENTS

We acknowledge funding from the Flemish Government (AI Research Program) and the Research Foundation - Flanders (FWO) through project number G0G2921N. We acknowledge the EuroHPC Joint Undertaking for awarding the project ID EHPC-BEN-2025B07-037, EHPC-BEN-2025B11-070 and EHPC-AIF-2025SC02-042 access to the EuroHPC supercomputer LEONARDO, hosted by CINECA (Italy) and the LEONARDO consortium. In addition, the resources and services used in this work were partially provided by the VSC (Flemish Supercomputer Center), funded by the Research Foundation - Flanders (FWO) and the Flemish Government.

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

# A APPENDIX

## A.1 USAGE OF LARGE LANGUAGE MODELS

We used a large language model (LLM) to polish the writing and, with assistance from the LLM, completed the plot visualizations.

## A.2 LIMITATIONS AND FUTURE WORKS

As part of future work, we plan to evaluate the performance of combining random Fourier encoding with the proposed binning-based approach. Furthermore, we aim to explore the convergence behavior of our method through the lens of the neural tangent kernel (NTK) framework (Jacot et al., 2018). This will help derive theoretical results on the training dynamics of our method.

## A.3 ROTATION VISUALIZATION

This section is a supplement to Section 4.2.

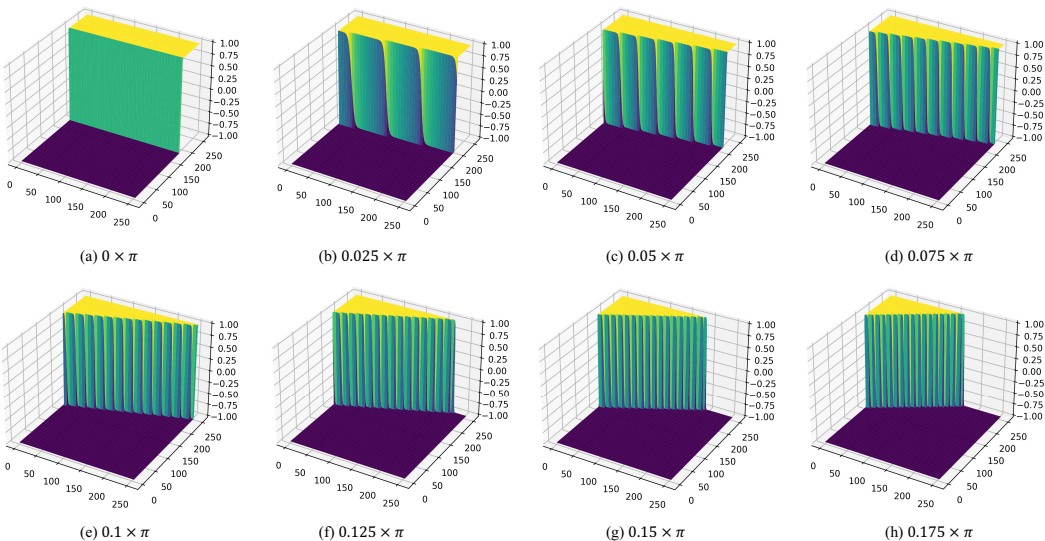

Figure 8: Rotation results of the two-dimensional hyperbolic tangent function at different angles.

## A.4 INITIALIZATION STRATEGY

This section is a supplement to Section 5.1 and Section A.8.

For the Neural network $f_{net}$ 's parameters in our method, we use the network initialization strategy from (Shen et al., 2022), which is the same as that used in (Sitzmann et al., 2020). For binning location parameters $\{u_i, v_i\}$ and side length parameters $\{h_i, w_i\}$, we adopt the initialization strategy of (Li et al., 2025), which was originally designed to determine the Gaussian function's center and length. This initialization provides a reasonable starting point for location and length parameter. Unlike (Li et al., 2025), however, our method does not assume a Gaussian formulation.

For the initialization of the rotation angle parameters $\theta_i$, we sample it from a normal distribution:

$$\theta_i \overset{\text{iid}}{\sim} \mathcal{N}(0, 0.05^2), \quad i = 1, 2, \ldots, M \tag{22}$$

For the initialization of the bin height $\lambda_i$, we sample it from uniform distribution:

$$\lambda_i \overset{\text{iid}}{\sim} U(0, 1), \quad i = 1, 2, \ldots, M \tag{23}$$

For steepness $k_i$, we initialize it according to the alternately selection strategy:

$$k_i = S[((i-1) \bmod L) + 1], \quad i = 1, 2, \ldots, M \tag{24}$$

For the CaCO$_3$ dataset, $S$ is set to $(600, 800)$ and $L$ is set to 2. For the Workpieces dataset, $S$ is set to $(25, 50, 75)$ and $L$ is set to 3. For the Medical-Axial dataset, $S$ is set to $(4, 6, 8, 12, 14, 16, 18, 20)$ and $L$ is set to 8. The $M$ above represents the number of bins. In our approach, we use $M=456$ for the CaCO$_3$ dataset and Workpiece dataset. For the Medical-Axial dataset, which is more complex and contains no rectangular structures, we increase $M$ to 3000.

For the initialization of the random Fourier coding in the comparison method, the element in RFC matrix is sampled independently from $\mathcal{N}(0, 4.0^2)$.

### A.5 LEARNING RATE SETTING

This section is a supplement to Section 5.1 and Section A.8.

For the CaCO$_3$ dataset, in the case of sparse reconstruction with 14 and 16 views, the learning rate ($lr$) for the neural network $f_{net}$ is set to $8.0 \times 10^{-4}$. The same $lr$ ($8.0 \times 10^{-4}$) is applied to the binning location parameters $\{u_i, v_i\}$ and side length $\{h_i, w_i\}$. The $lr$ for the rotation angle $\theta_i$ and the steepness $k_i$ are both set to $1.0 \times 10^{-4}$. The $lr$ for bin height $\lambda_i$ is set to $1.0 \times 10^{-5}$. For sparse reconstruction with 12 views, all learning rates remain the same as in the other view settings, except for the $lr$ of rotation angle $\theta_i$ is increased to $4.0 \times 10^{-4}$.

For the Workpiece dataset, in sparse reconstruction with 12 views, 14 views and 16 views, the $lr$ of the neural network $f_{net}$ is set to $7.0 \times 10^{-4}$. The $lr$ assigned to the binning location parameters $\{u_i, v_i\}$ is $8.0 \times 10^{-4}$. The $lr$ for the side length parameters $\{h_i, w_i\}$ is $4.0 \times 10^{-4}$. The $lr$ assigned to the rotation angle $\theta_i$ is specified as $8.0 \times 10^{-4}$. The $lr$ for the steepness $k_i$ is $4.0 \times 10^{-4}$. The $lr$ for the bin height $\lambda_i$ is set to $1.0 \times 10^{-5}$.

For the Medical-Axial dataset, in sparse reconstruction with 48 views, the $lr$ of the neural network $f_{net}$ is set to $2.0 \times 10^{-4}$. The $lr$ for the binning location parameters $\{u_i, v_i\}$ is $4.0 \times 10^{-4}$. The $lr$ for the side length parameters $\{h_i, w_i\}$ is $8.0 \times 10^{-4}$. The $lr$ assigned to the rotation angle $\theta_i$ is set to $1.0 \times 10^{-4}$. The $lr$ for the steepness $k_i$ is $4.0 \times 10^{-5}$. The $lr$ for the bin height $\lambda_i$ is $8.0 \times 10^{-4}$.

### A.6 EVALUATION METRICS

This section is a supplement to Section 5.2.

PSNR and SSIM are widely used to evaluate CT image quality in reconstruction and artifact removal (e.g., Liao et al., 2019; Reed et al., 2021; Shen et al., 2022; Xie & Blaschko, 2023; Liu et al., 2023). The PSNR values reported in Table 1 and Table 2 refers to the average PSNR (PSNR$_{avg}$) in CaCO$_3$ dataset and Workpieces dataset respectively. PSNR$_{avg}$ 's specific calculation formula is as follows:

$$\text{PSNR}_{\text{avg}} := \frac{1}{N_{object}} \sum_{i=1}^{N_{object}} \text{PSNR}_i \qquad (25)$$

where:

$$\text{PSNR}_i := 10 \log_{10}\left(\frac{\text{MAX}_I^2}{\frac{1}{HW}\sum_{x=1}^{H}\sum_{y=1}^{W}\left(I_i(x,y) - \hat{I}_i(x,y)\right)^2}\right), i = 1, 2, ..., N_{object} \qquad (26)$$

The $N_{object}$ indicates the number of objects to be reconstructed in the corresponding dataset. $H$ and $W$ represent the length and width of the object to be reconstructed, respectively. $I_i$ represents $i^{th}$ objects' ground truth, and $\hat{I}_i$ represents the reconstructed result of the $i^{th}$ object.

Similarly, the SSIM values reported in Table 1 and Table 2 refers to the average SSIM (SSIM$_{avg}$) in CaCO$_3$ dataset and Workpiece dataset respectively:

$$\text{SSIM}_{\text{avg}} := \frac{1}{N_{object}} \sum_{i=1}^{N_{object}} \text{SSIM}_i \qquad (27)$$

The specific calculation formula of $\text{SSIM}_i$ is as follows (Wang et al., 2004):

$$\text{SSIM}_i := \frac{(2\mu_{I,i}\mu_{\hat{I},i} + C_1)(2\sigma_{I\hat{I},i} + C_2)}{(\mu_{I,i}^2 + \mu_{\hat{I},i}^2 + C_1)(\sigma_{I,i}^2 + \sigma_{\hat{I},i}^2 + C_2)} \tag{28}$$

where:

$\mu_{I,i}$ : the mean values of $I_i$ over the $i^{th}$ object

$\mu_{\hat{I},i}$ : the mean values of $\hat{I}_i$ over the $i^{th}$ object

$\sigma_{I,i}^2$ : the variances of $I_i$ over the $i^{th}$ object

$\sigma_{\hat{I},i}^2$ : the variances of $\hat{I}_i$ over the $i^{th}$ object

$\sigma_{I\hat{I},i}$ : the covariance between $I_i$ and $\hat{I}_i$ over the $i^{th}$ object

$C_1, C_2$ : stabilization constants

For any $i \in \{1, 2, ..., N_{object}\}$, $\text{PSNR}_i$ was computed using the function `peak_signal_noise_ratio` from the `skimage.metrics` module in the `scikit-image` library (Van der Walt et al., 2014). Similarly, $\text{SSIM}_i$ was computed using the function `structural_similarity` from the same module.

### A.7  MORE VISUAL RESULTS ON CACO3 DATASET

This section is a supplement to Section 5.2.

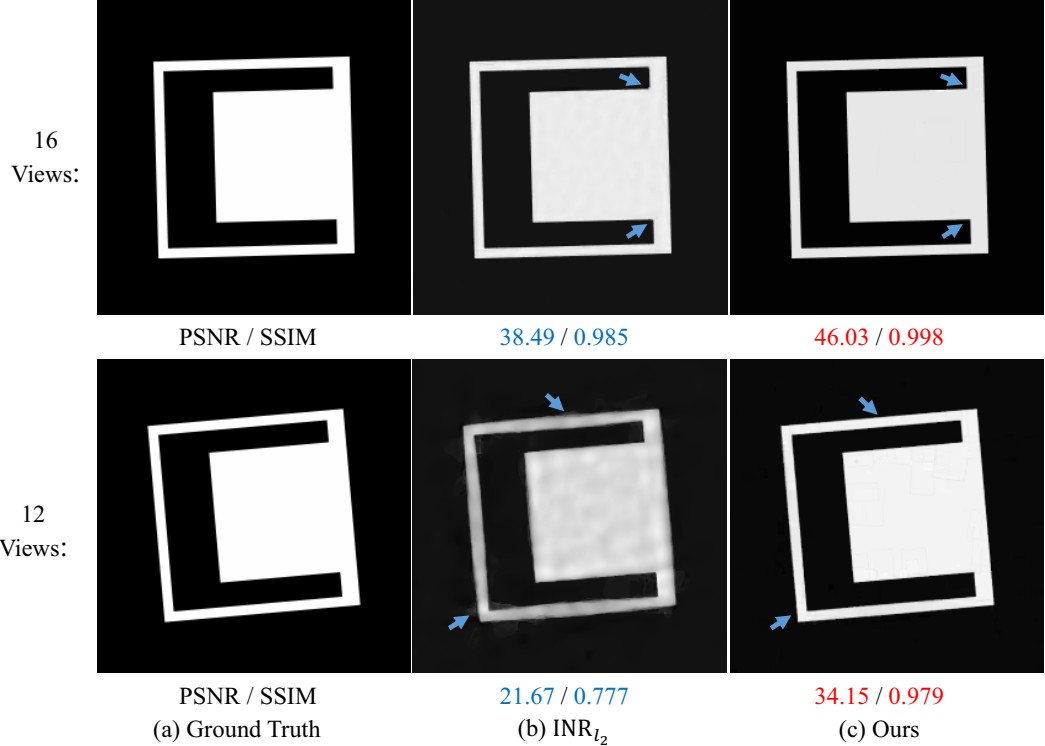

Figure 9: Reconstruction at 16 and 12 sparse views on the CaCO$_3$ dataset. The blue arrows show that the comparison method struggles to accurately reconstruct edges and corner points, whereas our method achieves better results. In addition, the comparison method exhibits wavy artifacts.

We performed a visual comparison on the CaCO$_3$ dataset under two boundary settings (12 and 16 views). As shown above, although the baseline $\text{INR}_{l_2}$ has four times more parameters than our method, it produces more wavy artifacts and less sharp boundaries and corners.

With 16 views, $\text{INR}_{l_2}$ achieves performance close to ours. However, after zooming in to reveal finer details, corner imperfections and wavy artifacts in flat regions remain visible. Although the comparison is not strictly fair (the neural network of $\text{INR}_{l_2}$ is larger than ours), it still highlights the advantage of our method.

## A.8 ROBUSTNESS ON MEDICAL DATASET

This section is a supplement to Section 5.1 and Section 5.2.

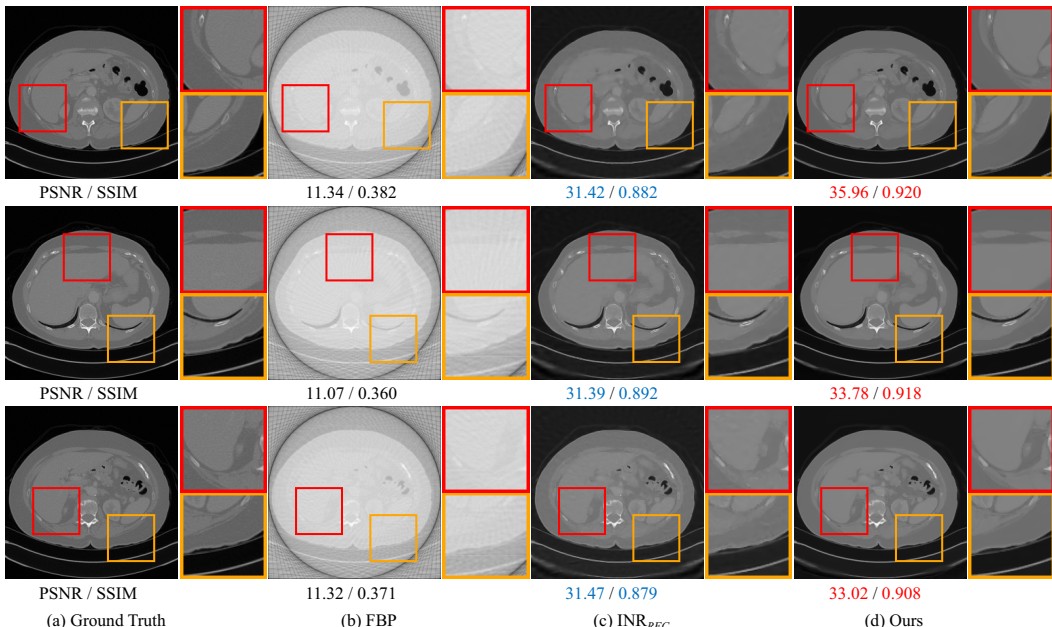

Figure 10: Reconstruction at 48 sparse views on the Medical-Axial dataset. The best numerical result is marked in red, and the second-best result is marked in blue. When **zooming** in, we observe that the comparison method $\text{INR}_{RFC}$ exhibits edge blurring and some non-edge areas are distorted by artifacts, whereas our method produces fewer artifacts and yields cleaner boundaries.

To verify the robustness of this work, we constructed a *Medical-Axial dataset* comprising 40 different slices, which are derived from the axial plane of the first four lung volumes in the training set of the Medical Segmentation Decathlon (Antonelli et al., 2022). Specifically, the four lung volumes are labeled Lung_001, Lung_003, Lung_004, and Lung_005 in the raw data. For each volume, we extracted 10 slices at 5-slice intervals starting from the 5th slice (i.e., the slices indexed at 5, 10, 15, ... 50). The projection of the *Medical-Axial dataset* is obtained from parallel-beam geometry with sparse view under 48 different angles.

To compare performance, we use INR based on RFC as baseline. For fair comparison, we ensure that the depth, width, activation function of the FCN, the length of the encoded output vector of the encoder, total number of training epochs, and the optimization method are exactly the same as those in our method. Because medical data contains more detailed textures than industrial data, we use a ten-layer FCN for both the comparative method and our approach. The initialization strategy and learning rate setting are described in Section A.4 and Section A.5. For simplicity, we abbreviate the comparative method as $\text{INR}_{RFC}$.

**Quantitative evaluation** For every method, Table 4 reports the average PSNR/SSIM on the different sub-datasets (10 slices from sub-dataset Lung_001, 10 slices from sub-dataset Lung_003, 10 slices from sub-dataset Lung_004, and 10 slices from sub-dataset Lung_005) and whole Medical-Axial dataset. We find that replacing random Fourier encoding with our NAB boosts $\text{INR}_{RFC}$ by 0.56 dB in whole Medical-Axial dataset.

Table 4: Reconstruction results on the whole Medical-Axial dataset and different sub-datasets.

| Methods | Lung_001 | | Lung_003 | | Lung_004 | | Lung_005 | | Whole dataset | |
|---|---|---|---|---|---|---|---|---|---|---|
| | PSNR↑ | SSIM↑ | PSNR↑ | SSIM↑ | PSNR↑ | SSIM↑ | PSNR↑ | SSIM↑ | PSNR↑ | SSIM↑ |
| INR$_{RFC}$ | 31.82 | 0.890 | 34.03 | 0.882 | 33.71 | 0.824 | 31.34 | 0.877 | 32.72 | 0.868 |
| Ours | **33.00** | **0.909** | **34.49** | **0.893** | **34.25** | **0.834** | **31.39** | **0.886** | **33.28** | **0.881** |

**Qualitative evaluation** The qualitative evaluation of reconstruction is shown in Figure 10. We found that our method achieves good performance even in medical data with complex shapes that do not contain rectangular objects, and our results do not show wave-like artifacts. This demonstrates that our method is robust on medical dataset.

