# OpenReview forum: "NAB: Neural Adaptive Binning for Sparse-View CT reconstruction"
_ICLR.cc/2026/Conference — ICLR 2026 Poster_

### Official Review · Reviewer_J6rB · 2025-10-30

**Soundness:** 3
**Presentation:** 3
**Contribution:** 3
**Rating:** 4
**Confidence:** 4

**Summary:**

The paper offers a clear shape-aware alternative to RFC-based INRs by constructing trainable, rotated bins that better match industrial rectangular priors. Instead of random Fourier features in INRs, NAB encodes each spatial coordinate via differentiable rectangular bins formed by differences of shifted tanh functions along two axes, with learnable translation, scale, rotation, height, and steepness. A small FCN maps these bin features to attenuation values, and training minimizes projection-domain MSE with a differentiable forward operator. The authors analyze a limiting property showing convergence of the soft bins toward hard binary partitions as steepness increases. Experiments report sizeable PSNR/SSIM gains over DIP, Wire, and several INR variants, plus ablations on bin parameters.

**Strengths:**

- **Shape-aware coordinate encoding.**
  The paper proposes a shape-aware coordinate encoding scheme tailored to the geometric characteristics of industrial components, effectively leveraging rectangular priors common in real-world applications. For non-canonical rectangles, the model introduces learnable rotation parameters to handle arbitrary orientations, providing greater flexibility. The integration of **multi-scale steepness** further extends the representation from sharp-edged rectangles to smooth, bulge-like bins, enhancing expressiveness.

- **Novel prior design for INR tasks.**
  Incorporating a shape-aware prior into the coordinate encoding stage of an implicit neural representation (INR) is both meaningful and underexplored. This design introduces a new perspective for geometry-adaptive representation learning and offers potential inspiration for future INR-based industrial CT research.

**Weaknesses:**

- **Limited dataset diversity and realism.**
  The evaluation is conducted on only two simulated datasets—**CaCO₃** and **Workpieces**—both of which are relatively simple. The CaCO₃ dataset lacks geometric complexity, and the Workpieces dataset includes only a small number of slices. It would strengthen the paper to include experiments on more complex real-world components and on real projection data.

- **Narrow baseline selection.**
  The comparative baselines are limited to several RFC-based MLP variants. Given that this paper focuses on a specific industrial CT application, it would be more convincing to include comparisons with **advanced INR methods** such as **SIREN** [1] and **InstantNGP** [2], which are now standard benchmarks for implicit representation learning.
---
Overall, the paper presents an interesting idea with potential impact. However, the experimental validation is not sufficiently thorough. The choice of baselines lacks persuasiveness, and the datasets used appear overly simplistic. Providing more rigorous and comprehensive experiments would considerably strengthen the paper and make the results more convincing. I would be open to reconsidering my assessment if the authors can present more substantial empirical evidence in a future revision.

**Questions:**

- **On initialization of the neural network.**
  The appendix states: “For the neural network $f_{\text{net}}$’s parameters, we use the initialization strategy from (Shen et al., 2022).” This is unclear. In NeRP, initialization encodes temporal similarity of the same object. How is this strategy adapted in the current paper? Please clarify how the initialization is performed here.

- **On relation to 3D Gaussian representations.**
  The proposed shape-aware rectangular encoding appears conceptually similar to explicit 3D Gaussian representations, except that rectangles replace Gaussian ellipsoids. Could the authors add **3D Gaussian representation** as a baseline to directly compare their relative performance?

---

> ### Author Response · Authors · 2025-11-25
> **Response to Reviewer  J6rB (Weaknesses 1)**
>
> **We sincerely thank Reviewer J6rB for their comments and for recognizing the interesting ideas and potential impact of our work. Their comments offer valuable perspectives that will help us further strengthen this work.  We add additional dataset and baselines.**
>
> >**Weaknesses 1:**
> > “Limited dataset diversity and realism.The evaluation is conducted on only two simulated datasets—CaCO₃ and Workpieces—both .…”
>
> >**Response:**
>
> We have added Section A.7 in the Appendix, where we provide both visualization and numerical results for medical reconstruction.**  Please refer to the revised paper in the OpenReview system.
>
> Specifically, we constructed a Medical-Axial dataset comprising 40 different slices, which are derived from the axial plane of the first four lung volumes in the training set of the Medical Segmentation Decathlon[1]. Specifically, the four lung volumes are labeled Lung\_001, Lung\_003, Lung\_004, and Lung\_005 in the raw data. For each volume, we extracted 10 slices at 5-slice intervals starting from the 5th slice (i.e., the slices indexed 5, 10, 15, ... 50). The projection in the dataset is obtained from parallel-beam geometry with 48 sparse views.
>
> To compare performance, we use INR based on RFC as baseline. For fair comparison, we ensure that the depth, width, activation function of the FCN, the length of the encoded output vector of the encoder, total number of training epochs, and the optimization method are exactly the same as those in our method.  Because medical data contains more detailed textures than industrial data, we use a ten-layer FCN for both the comparative method and our approach.
> For simplicity, we abbreviate the comparative method as $INR_ {RFC}$.
>
> Table 1: Reconstruction results on the whole Medical-Axial dataset and different sub-datasets.
>
> | Methods        | Lung_001 PSNR | Lung_001 SSIM | Lung_003 PSNR | Lung_003 SSIM | Lung_004 PSNR | Lung_004 SSIM | Lung_005 PSNR | Lung_005 SSIM | Whole dataset PSNR | Whole dataset SSIM |
> |----------------|---------------|---------------|---------------|---------------|----------------|----------------|-----------------|----------------|---------------------|---------------------|
> | $INR_{RFC}$    | 31.82         | 0.890         | 34.03         | 0.882         | 33.71          | 0.824          | 31.34           | 0.877          | 32.72              | 0.868              |
> | **Ours**       | **33.00**     | **0.909**     | **34.49**     | **0.893**     | **34.25**       | **0.834**       | **31.39**       | **0.886**       | **33.28**          | **0.881**          |
>
>
> **Quantitative evaluation** Table 1 reports the average PSNR/SSIM on the different sub-datasets (10 slices from sub-dataset Lung\_001, 10 slices from sub-dataset Lung\_003, 10 slices from sub-dataset Lung\_004, and 10 slices from sub-dataset Lung\_005) and whole Medical-Axial dataset. We find that **replacing random Fourier encoding with our NAB boosts $INR_ {RFC}$ by 0.56 dB in whole Medical-Axial dataset**.
>
> **Qualitative evaluation** The qualitative evaluation of reconstruction is shown in Figure 9 of the revised paper. We found that **our method achieves good performance even in medical data with complex shapes that do not contain rectangular objects, and our results do not show wave-like artifacts.
>
> **Before incorporating the medical data, the first manuscript included a total of 29 industrial objects. Every object to be reconstructed is a two-dimensional phantom (although our method can be naturally extended to 3D by replacing the 2D coordinates in the experiments with 3D coordinates). We would also like to point out that recent literature [2] performed algorithmic verification using 18 phantoms, which we believe that our experimental scale is reasonable.**
>
> The use of 2D phantoms is a widely accepted setting in the community [3–6].  Moreover, 2D CT scanning can be implemented in actual CT systems (When collecting experimental data, it is only need to count X-rays parallel to the horizontal plane).
>
> [1] Antonelli M, Reinke A, Bakas S, et al. The medical segmentation decathlon[J]. Nature communications, 2022, 13(1): 4128.
>
> [2] Zha, Ruyi, et al. "R2-Gaussian: rectifying radiative Gaussian splatting for tomographic reconstruction." Proceedings of the 38th International Conference on Neural Information Processing Systems. 2024.
>
> [3] Jin, Kyong Hwan, et al. "Deep convolutional neural network for inverse problems in imaging." IEEE transactions on image processing 26.9 (2017): 4509-4522.
>
> [4] Song, Yang, et al. "Solving Inverse Problems in Medical Imaging with Score-Based Generative Models." International Conference on Learning Representations.
>
> [5] Wu, Qing, et al. "Self-supervised coordinate projection network for sparse-view computed tomography." IEEE Transactions on Computational Imaging 9 (2023): 517-529.
>
> [6] Shi, Jiayang, et al. "Implicit Neural Representations for Robust Joint Sparse-View CT Reconstruction." Transactions on Machine Learning Research.

---

> ### Author Response · Authors · 2025-11-25
> **Response to Reviewer J6rB (Weaknesses 2)**
>
> >**Weaknesses 2:**
> > “Narrow baseline selection.The comparative baselines are limited to several RFC-based MLP variants. Given that this paper focuses on a specific industrial CT application.…”
>
> >**Response:**
>
> We added SIREN[1] and Instant-NGP[2] as new baselines. For fair comparison, we ensured that the depth, width, and total number of training epochs of the FCN, as well as the optimization method, are exactly the same as in our method.  **From the table, overall, our method outperforms very well.**
>
> Instant-NGP has ten times the number of parameters of our method because the hash encoding part has many learnable parameters. This is an inherent limitation of the Instant-NGP method, which is also criticized in the literature[3]:" imposing an additional grid structure on neural fields creates numerous additional hyperparameters." Beyond the excessively large number of parameters in the hash-encoding component, its interpretability is also somewhat limited. the literature [3] further states that Instant-NGP lacks a theoretical explanation for the effectiveness of its hash-grid structure.
>
> Their performance on the CaCO3 and Workpieces datasets is shown in Tables 1 and 2. We found that SIREN performed particularly poorly on the CaCO3 dataset. We posit that this issue arises because the Sine activation function used in the FCN of SIREN is inherently ill-suited for representing rectangular structures. Furthermore, in our previous experiments, we also found that SIREN performs worse than simply using ReLU as the activation function for CT reconstruction of objects that are piecewise continuous. We also discovered an interesting phenomenon: SIREN's PSNR on the Workpieces dataset is similar to Instant-NGP, but its SSIM is much lower. We attribute this to the fact that SIREN produces a large number of wave-like artifacts (discovered through visualization) — which aligns with the motivation of this work, as trigonometric function(Although this paper only refutes the trigonometric functions used in the encoding stage) inherently generate wave-like artifacts when encountering objects with rectangular structures.
>
> Table 1: Numerical results of reconstruction on the CaCO3 dataset.
>
> | Methods     | 16-view PSNR | 16-view SSIM | 14-view PSNR | 14-view SSIM | 12-view PSNR | 12-view SSIM | Trainable Params |
> |-------------|--------------|--------------|--------------|--------------|--------------|--------------|------------------|
> | Instant-NGP | 30.81        | 0.953        | 30.03        | 0.946        | 27.23        | **0.911**        | 2.96×10⁶         |
> | SIREN       | 19.82        | 0.227        | 18.34        | 0.193        | 16.98        | 0.164        | 2.49×10⁵         |
> |  $INR_{f}$         | 29.01        | 0.934        | 27.10        | 0.891        | 25.08        | 0.854        | 2.49×10⁵         |
> | Ours        | **43.61**    | **0.996**    | **40.07**    | **0.977**    | **34.72**    |  0.888    | 2.52×10⁵         |
>
> Table 2: Numerical results of reconstruction on the Workpieces dataset.
>
> | Methods     | 16-view PSNR | 16-view SSIM | 14-view PSNR | 14-view SSIM | 12-view PSNR | 12-view SSIM | Trainable Params |
> |-------------|--------------|--------------|--------------|--------------|--------------|--------------|------------------|
> | Instant-NGP | 31.17        | 0.904        | 31.95        | 0.914        | 28.38        | 0.832        | 2.96×10⁶         |
> | SIREN       | 31.56        | 0.614        | 30.03        | 0.541        | 29.11        | 0.564        | 2.49×10⁵         |
> | $INR_{f}$      | 33.34        | 0.911        | 34.03        | 0.912        | 27.37        | 0.787        | 2.49×10⁵         |
> | Ours        | **36.26**    | **0.938**    | **35.23**    | **0.941**    | **32.76**    | **0.909**    | 2.52×10⁵         |
>
> [1] Sitzmann, Vincent, et al. "Implicit neural representations with periodic activation functions." Advances in neural information processing systems 33 (2020): 7462-7473.
>
> [2] Müller, Thomas, et al. "Instant neural graphics primitives with a multiresolution hash encoding." ACM transactions on graphics (TOG) 41.4 (2022): 1-15.
>
> [3] Luo, Steven Tin Sui. "A New Perspective To Understanding Multi-resolution Hash Encoding For Neural Fields." arXiv preprint arXiv:2505.03042 (2025).

---

> ### Author Response · Authors · 2025-11-25
> **Response to Reviewer J6rB (Questions 1)**
>
> >**Questions 1:**
> > “On initialization of the neural network.The appendix states: “For the neural network $f_{\text{net}}$’s parameters, we use the initialization strate...”
>
> >**Response:**
>
> We thank the reviewers for pointing out our inaccurate description. **We did not utilize any temporal similarity**. We simply adopted the FCN' s initialization strategy of the classic baseline(classic INR without a priori) mentioned in reference [1]. Specifically, the initialization of FCN in our method is sampling from a uniform distribution.
>
> **We want to reiterate that the literature[1] embeds the neural network parameters from the previous time step into the next time step, which provides the FCN with prior information. Our FCN's initialization, however, does not use any prior information.**
>
> [1] Shen L, Pauly J, Xing L. NeRP: implicit neural representation learning with prior embedding for sparsely sampled image reconstruction[J]. IEEE Transactions on Neural Networks and Learning Systems, 2022, 35(1): 770-782.

---

> ### Author Response · Authors · 2025-11-25
> **Response to Reviewer J6rB (Questions 2)**
>
> >**Questions 2:**
> > “On relation to 3D Gaussian representations. The proposed shape-aware rectangular encoding appears conceptually similar to explicit 3D Gaussian representations, except that rectangles replace .…”
>
> >**Response:**
>
> **Thank you for your comment. However, from the current wording of the review, it is unclear what specific “3D Gaussian representation” the reviewer is referring to?  Because I didn't find any references in the reviewers' comments.**
>
> Specifically, I'm unsure whether "3D Gaussian representation" refers to a specific technique or Encoding method using a 3D Gaussian function.
>
> (a) If reviewer refers “3D Gaussian representation”  to INR encoding using a 3D Gaussian function, based on my current knowledge,  there's no work in the CT reconstruction field.
>
> (b). If reviewer refers ”3D Gaussian representation"  to the 3D Gaussian Splatting technique, there is indeed relevant literature [1, 2, 3].
>
> If it is situation (b) above.  We want to first distinguish the main differences between 3D Gaussian Splatting and our work:
>
> (1) 3D Gaussian Splatting does not use neural networks, whereas our work use neural networks.
>
> (2) The 2D form of a Gaussian function is bell-shaped, while the 2D form of the basis functions in our method is rectangular. Although our basis functions can be generalized to bell shapes (Figure 4 in the paper), the Gaussian function cannot be generalized to rectangular (plateau-shaped) forms. The basis functions in our method are important for reconstructing rectangular regions in industrial objects, where bell shapes are not suitable. Similarly, in the 3D form, the basis functions of this work will become a solid cuboid (which can be transformed into a general hexahedron by performing differentiable rotations on each face, and degenerate into an ellipsoid by changing the steepness of each face). Conversely, 3D Gaussians  can only represent ellipsoids, not cuboids.
>
> (3) The mathematical form of the basis functions in our work is entirely different from that of 3D Gaussian.
>
> However, we will try our best to complete the comparative experiments of 3D Gaussian Splatting, this allows us to illustrate the problem numerically(If it is situation (b) above).
>
> [1] Kerbl, Bernhard, et al. "3D Gaussian splatting for real-time radiance field rendering." ACM Trans. Graph. 42.4 (2023): 139-1.
>
> [2] Zha, Ruyi, et al. "R2-Gaussian: rectifying radiative Gaussian splatting for tomographic reconstruction." Proceedings of the 38th International Conference on Neural Information Processing Systems. 2024.
>
> [3] Li, Yingtai, et al. "3DGR-CT: Sparse-view CT reconstruction with a 3D Gaussian representation." Medical Image Analysis (2025): 103585.

---

> ### Author Response · Authors · 2025-11-26
>
> I hope the above response has addressed the reviewer’s current concerns. If anything is unclear or if there are any mistakes in our description, please feel free to correct us, and we will do our best to make the necessary revisions. We also hope that the reviewer will feel free to contact us  if there are any additional concerns.

---

> ### Author Response · Authors · 2025-12-02
> **Additional Response to Reviewer J6rB (Questions 2) — Experimental data of the Gaussian representation-based method**
>
> Although our response clearly outlines the fundamental differences between our method and Gaussian representation:
>
> **1. The mathematical modeling of our method is completely different to Gaussian representation.**
>
> **2. Our method employs neural networks whereas Gaussian representation does not.**
>
> **3. Our method offers broader applicability, especially for industrial products.**
>
> **We have also conducted numerical experiments to further demonstrate these differences quantitatively:**
>
> Table 3: Numerical results of reconstruction on the CaCO3 dataset.
>
> | Methods                    | 16-view PSNR | 16-view SSIM | 14-view PSNR | 14-view SSIM | 12-view PSNR | 12-view SSIM |
> |----------------------------|--------------|--------------|--------------|--------------|--------------|--------------|
> | Gaussian Representation   | 21.35        | 0.800        | 20.74        | 0.782        | 20.27        | 0.771        |
> | Ours                       | **43.61**    | **0.996**    | **40.07**    | **0.977**    | **34.72**    | **0.888**    |
>
> The above experiments demonstrate that, when dealing with industrial objects filled with rectangles, the numerical performance of Gaussian representation is inferior to our method.
>
> We believe there are two reasons behind this:
>
> 1. The basis functions in Gaussian representations are all Gaussian and thus elliptical, making them less effective when handling straight-line edges.
>
> 2. Gaussian representation does not have a neural network, therefore its representational power is worse than that of representations with a neural network.

---

### Official Review · Reviewer_9B2M · 2025-10-30

**Soundness:** 3
**Presentation:** 3
**Contribution:** 3
**Rating:** 4
**Confidence:** 5

**Summary:**

This work proposes a new reconstruction method based on implicit neural representations (INR), namely NAB, for industrial CT imaging. The main motivation is that most man-made industrial objects are composed of rectangular structures, while existing INR methods fail to exploit such explicit shape priors, leading to suboptimal performance. To address this issue, the proposed NAB introduces a new encoding strategy, where a hyperbolic tangent function is used to form an adaptive binning operation. With this encoding, the shape prior can be implicitly incorporated into the network optimization, resulting in improved reconstruction quality. The experiments also demonstrate the superiority of the proposed NAB over existing INR methods.

**Strengths:**

- The key motivation and idea of this paper are interesting. Incorporating shape priors is a reasonable way to improve the reconstruction accuracy for industrial CT imaging.
- The proposed encoding strategy (i.e., the adaptive binning operation) is technically sound.
- The paper is clearly written and easy to follow.

**Weaknesses:**

My main concern lies in the experimental evaluation:
- The compared baselines mainly include the random Fourier encoding, but exclude Instant-NGP [1]. To my knowledge, Instant-NGP is much more powerful than random Fourier encoding in various inverse problems, such as CT or MRI.
- In Table 1, the $INR_{l_1}$ performs worse than $INR_f$, which is a bit strange since the former has more learnable parameters.
- The random Fourier encoding has two key hyperparameters, the mean and the standard deviation, which strongly affect its learning capacity. However, these hyperparameters are not reported in the paper.
- From Figures 5 and 6, the objects from the Workpieces dataset have more complex shapes than those from the CaCO3 dataset. However, the improvement achieved by NAB on the Workpieces dataset is relatively small. Does this indicate that NAB does not perform well on more complex objects?
- In Table 2, the results of $INR_{l_2}$ are missing.
- In Figures 5 and 6, the qualitative results of the second-best method ($INR_{l_2}$) are also missing.

[1] Müller, Thomas, et al. "Instant neural graphics primitives with a multiresolution hash encoding." ACM transactions on graphics (TOG) 41.4 (2022): 1-15.

**Questions:**

Overall, this paper is interesting in its idea and motivation, but the current experimental evaluation is somewhat limited. Therefore, I cannot recommend acceptance in the current version. I would be willing to raise my rating if the above concerns can be properly addressed.

---

> ### Author Response · Authors · 2025-11-24
> **Response to Reviewer 9B2M (Weaknesses 1)**
>
> **We sincerely thank Reviewer 9B2M for their constructive comments, which provide guidance for further improving this work. We also want to sincerely thank Reviewer 9B2M for appreciating the interesting idea and motivation of our paper, which highlights the novelty of our work. Regarding the questions, we have incorporated additional baseline and provide detailed responses below.**
>
> >**Weaknesses 1:**
> > “The compared baselines mainly include the random Fourier encoding, but exclude Instant-NGP [1]. To my knowledge, Instant-NGP is much more powerful than random ...”
>
> >**Response:**
>
> We appreciate Reviewer 9B2M for the valuable reminder to include Instant-NGP [1,2], which is important for method comparison. The fundamental difference between  Instant-NGP and $INR_ f$  is that Instant-NGP does not use random Fourier encoding for coordinates, but instead uses a hash encoding.
>
> For fair comparison, we ensure that the depth, width, activation function of the FCN, as well as the total number of training epochs and the optimization method, are exactly the same as in our method. The results are shown in Tables 1 and Tables 2. **On the CaCO₃ dataset, our method outperforms Instant-NGP by 12.8 dB, 10.04 dB, and 7.49 dB at 16, 14, and 12 views**, respectively. On the Workpieces dataset, our **method outperforms Instant-NGP by 5.09 dB, 3.28 dB, and 4.38 dB at 16, 14, and 12 views**, respectively.
>
> In addition, as shown in Table 1 and Table2, **Instant-NGP has ten times the number of parameters of our method because the hash encoding part has many learnable parameters**. This is an inherent limitation of the Instant-NGP method, which is also criticized in the literature[3]:" *imposing an additional grid structure on neural fields creates numerous additional hyperparameters.*"
>
> Beyond the excessively large number of parameters in the hash-encoding component, its interpretability is also somewhat limited. the literature [3] further states that **Instant-NGP lacks a theoretical explanation for the effectiveness of its hash-grid structure. By contrast, as noted by Reviewer 4f2T, our method exhibits strict mathematical interpretability.**
>
> Table 1: Reconstruction results on the CaCO₃ dataset.
>
> | Methods     | 16-view PSNR | 16-view SSIM | 14-view PSNR | 14-view SSIM | 12-view PSNR | 12-view SSIM | Trainable Params |
> |-------------|--------------|--------------|--------------|--------------|--------------|--------------|------------------|
> | Instant-NGP | 30.81        | 0.953        | 30.03        | 0.946        | 27.23        | **0.911**    | 2.96×10⁶     |
> | $INR_ f$        | 29.01        | 0.934        | 27.10        | 0.891        | 25.08        | 0.854        | 2.49×10⁵         |
> | Ours        | **43.61**    | **0.996**    | **40.07**    | **0.977**    | **34.72**    | 0.888        | 2.52×10⁵         |
>
> Table 2: Reconstruction results on the Workpieces dataset.
>
> | Methods     | 16-view PSNR | 16-view SSIM | 14-view PSNR | 14-view SSIM | 12-view PSNR | 12-view SSIM | Trainable Params |
> |-------------|--------------|--------------|--------------|--------------|--------------|--------------|------------------|
> | Instant-NGP | 31.17        | 0.904        | 31.95        | 0.914        | 28.38        | 0.832        | 2.96×10⁶     |
> | $INR_ f$         | 33.34        | 0.911        | 34.03        | 0.912        | 27.37        | 0.787        | 2.49×10⁵         |
> | Ours        | **36.26**    | **0.938**    | **35.23**    | **0.941**    | **32.76**    | **0.909**    | 2.52×10⁵         |
>
> [1] Müller, Thomas, et al. "Instant neural graphics primitives with a multiresolution hash encoding." ACM transactions on graphics (TOG) 41.4 (2022): 1-15.
>
> [2] Zha R, Zhang Y, Li H. NAF: neural attenuation fields for sparse-view CBCT reconstruction[C]//International Conference on Medical Image Computing and Computer-Assisted Intervention. Cham: Springer Nature Switzerland, 2022: 442-452.
>
> [3] Luo S T S. A New Perspective To Understanding Multi-resolution Hash Encoding For Neural Fields[J]. arXiv preprint arXiv:2505.03042, 2025.

---

> ### Author Response · Authors · 2025-11-24
> **Response to Reviewer 9B2M (Weaknesses 2)**
>
> >**Weaknesses 2:**
> > “In Table 1, the $INR_{l_1}$ performs worse than $INR_f$, which is a bit strange since the former has more learnable parameters. ...”
>
> >**Response:**
>
> We also observed this "Performance-parameter reversibility"  in our experiments. Our reasoning is as follows: For a given dataset and an RFC-based INR, slightly increase the parameters does not guarantee better performance. This is related to the dataset characteristic and the structure of the $INR$.
>
> The CaCO₃ dataset is composed entirely of rectangular structures, a characteristic that further amplifies the instability of RFC-based INR. Although $INR_ {l_1}$ has slightly more parameters than both $INR_ f$ and our method, $INR_ {l_1}$ is worse than $INR_ f$  across all views. Due to the challenging characteristics of this dataset, we included an unfair comparison method $INR_ {l_ 2}$ on CaCO₃ dataset,  whose parameters are four times larger than ours.
>
> Correspondingly, on the Workpieces dataset which includes non-rectangular structures, the larger, parameter-rich $INR_{l_1}$ consistently outperforms $INR_ f$ across all views.

---

> ### Author Response · Authors · 2025-11-24
> **Response to Reviewer 9B2M (Weaknesses 3, Weaknesses 4)**
>
> >**Weaknesses 3:**
> > “The random Fourier encoding has two key hyperparameters, the mean and the standard deviation, which strongly affect its learning capacity. However, these hyperparameters are not reported in the paper. ...”
>
> >**Response:**
>
> The mean of the normal distribution which is used to construct the RFC frequency matrix is 0, the standard deviation of the normal distribution is 4. According to the literature[1], the elements in RFC frequency matrix are always sampled from a normal distribution with a mean of 0. We will add this information to the revised paper.
>
> >**Weaknesses 4:**
> From Figures 5 and 6, the objects from the Workpieces dataset have more complex shapes than those from the CaCO3 dataset. However, the improvement achieved by NAB ...*
>
> >**Response:**
>
> Because the workpieces contain curved shapes, and this work is highly adapted to rectangular shapes, the curvature is a generalization of this work. As described in Section 4.5 SHAPE GENERALIZATION: "*We further validate the effectiveness of the multi-scale mechanism in Section 5, which involves a dataset containing regions with non-zero curvature*".  Therefore, the performance improvement on datasets containing curved objects is less than that on datasets containing only rectangular objects, which is consistent with the mathematical model of the encoding.
>
> However, we would like to emphasize that **for industrial objects that contain both rectangular and curved components, our method still outperforms approaches based on random Fourier encoding and hash encoding**.
>
> **In addition, we have included a new medical dataset. Please refer to Appendix A.7 of the revised manuscript on the OpenReview system**. The shapes in this medical dataset are more complex, yet our method still outperforms RFC-based approaches. This demonstrates that our method can effectively handle complex shapes (Although it cannot achieve the exceptionally high performance it attains on industrial datasets composed entirely of rectangular objects).
>
> We currently believe that the **good performance of our method come from its inherent suitability for piecewise-continuous objects, which include both industrial items and organs**.
>
> [1] Tancik M, Srinivasan P, Mildenhall B, et al. Fourier features let networks learn high frequency functions in low dimensional domains[J]. Advances in neural information processing systems, 2020, 33: 7537-7547.

---

> ### Author Response · Authors · 2025-11-24
> **Response to Reviewer 9B2M (Weaknesses 5, Weaknesses 6)**
>
> >**Weaknesses 5:**
> > “In Table 2, the results of $INR_{l_2}$ are missing ...”
>
> >**Response:**
>
> We would like to clarify that introducing $INR_ {l_ 2}$ for the CaCO₃ dataset was not intended as an fair comparison. **Because $INR_ {l_ 2}$ employs a deeper FCN architecture (7 layers) compared with our 4-layer network, resulting in roughly four times more parameters**. However, we insist on making this unfair comparison on CaCO₃ dataset for the following reasons:
>
> 1.  The CaCO₃ dataset is composed entirely of rectangular structures, a characteristic that further amplifies the instability of RFC-based INR. Although $INR_ {l_1}$ has slightly more parameters than both $INR_ f$ and our method, $INR_ {l_1}$ is worse than $INR_ f$ across all views.
> 2. The CaCO₃ dataset consists entirely of rectangular shapes, the baseline methods $INR_ {f}$ and $INR_ {l_ 1}$ exhibit extremely limited performance on it.
>
> However, the Workpieces dataset do not exhibit the two characteristics described above. Therefore, in the Workpieces dataset, we did not include such an unfair comparison with $INR_ {l_2}$. So, in paper‘s Table 2, the results of $INR_{l_ 2}$ are missing.
>
> We also stated the reason why we insist on making this unfair comparison on CaCO₃ dataset in Section 5.2 PERFORMANCE COMPARISON of the original submission: "*Since the compared methods exhibit limited performance on the CaCO₃ dataset, we additionally evaluate $INR_ {l_2}$, an INR based on a seven-layer FCN in which each hidden layer has 456 neurons. Notably, $INR_{l_ 2}$  has four times more parameters than our method.*"
>
> We hope this clarification helps address the concern. Nevertheless, if the reviewer considers this explanation insufficient, we would be happy to do a further explanation.
>
>
> >**Weaknesses 6:**
> In Figures 5 and 6, the qualitative results of the second-best method ($INR_ {l_ 2}$) are also missing...*
>
> >**Response:**
>
> **Since Figure 5 presents the results on the CaCO₃ dataset, we will add the $INR_ {l_ 2}$ 's reconstruction visualization to maintain consistency with Table 1.**
>
> Regarding Figure 6, which reports the results on the Workpieces dataset, the situation is similar to that described in Weaknesses 5.
>
> **Finally, we would like to highlight that this paper draws attention to the often-overlooked encoding component and proposes constructing basis functions from the perspective of function approximation theory. By replacing the encoding with appropriate basis functions, our method achieves full mathematical interpretability while maintaining strong empirical performance. We hope this perspective can inspire future research directions, such as exploring how different choices of basis functions shape the output image manifold and what type of manifolds are best suited for CT reconstruction tasks.**

---

> ### Author Response · Authors · 2025-11-26
>
> I hope the above response has addressed the reviewer’s current concerns. If anything is unclear or if there are any mistakes in our description, please feel free to correct us, and we will do our best to make the necessary revisions. We also hope that the reviewer will feel free to contact us  if there are any additional concerns.

---

### Official Review · Reviewer_4f2T · 2025-10-31

**Soundness:** 3
**Presentation:** 3
**Contribution:** 3
**Rating:** 4
**Confidence:** 5

**Summary:**

The paper proposes a neural adaptive binning scheme as an alternative to traditional random Fourier feature coordinate encoding. The scheme not only supports rigid body transformations but also achieves scaling transformations, which enables explicit modeling of the rectangular priors commonly encountered in industrial CT scenarios.

**Strengths:**

The proposed Neural Adaptive Binning (NAB) method exhibits strong mathematical interpretability, which provides theoretical guarantees for rectangular priors.

**Weaknesses:**

The numerical results of the proposed method do not demonstrate irreplaceability, as comparable effects could be achieved through regularization techniques. Additionally, there is a lack of validation in specialized application scenarios.

**Questions:**

1. The authors mention that implicit neural representations with random Fourier coding often produce distinctive wave-like artifacts during reconstruction. However, upon comparing the INR results with the proposed method in Figures 5 and 6, the differences in reconstruction outcomes primarily manifest in the internal smoothness rather than at the edges. In practice, such artifacts can often be mitigated through regularization techniques. The authors are requested to clarify the advantages of their proposed method compared to alternative approaches, such as incorporating Total Variation (TV) regularization [1] into the network architecture.

[1] Deep Convolutional Neural Networks with Spatial Regularization, Volume and Star-Shape Priors for Image Segmentation. J. Math. Imaging Vis. 64(6): 625-645 (2022)

2. The results in Table 2 appear unusual, as the INR method shows better performance with 14 views than with 16 views. It seems counterintuitive and requires explanation.

3. The reconstruction results of INR in Figure 6 appear excessively smooth. Could this indicate potential overfitting to the training data?

4. The evaluation on medical datasets should be conducted to validate the robustness.

---

> ### Author Response · Authors · 2025-11-24
> **Response to Reviewer 4f2T (Questions 1)**
>
> **We thank Reviewer 4f2T for their thoughtful comments, which will help us further improve this work. We also appreciate Reviewer‘s recognition of the strong mathematical interpretability of our method. Below, we respond to each of the reviewer’s points in turn.**
>
> >**Question 1:**
> > “The authors mention that implicit neural representations with random Fourier coding often produce distinctive wave-like artifacts during reconstruction. However, upon comparing the INR…”
>
> >**Response:**
>
> We report here experiments with adding a $ \text{TV loss} $  in the image domain, which can be regarded as a prior on the image to be reconstructed. However, even with the $ \text{TV loss} $ and after tuning the coefficient before the TV term, the performance still falls far short of our method.
>
> We present the reconstruction results of $INR_ f$ under $ \text{MSE loss}$  and $ \text{TV loss} $ with different coefficients, as summarized in Table 1 below.  During training, we always compute the MSE loss in the projection domain, and use $ \text{MSE}_ \text{projection domain}$ + $\lambda$ · $ \text{TV loss}_ \text{image domain} $ as the final loss. We can observe that when $\lambda$ equals $10^{-2}$, the performance is the best. However, for the 16, 14, and 12 view settings, the results with $10^{-2} \times \text{TV loss}$ are **8.9 dB**, **7.28 dB**, and **5.73 dB** **lower than ours**, respectively. These are substantial differences.
>
> Table 1. Numerical results of reconstruction on the CaCO$_ 3$ dataset.
>
> | Methods               | 16-view PSNR | 16-view SSIM | 14-view PSNR | 14-view SSIM | 12-view PSNR | 12-view SSIM |
> |:---------------------:|:------------:|:------------:|:------------:|:------------:|:------------:|:------------:|
> | $ \text{MSE} + 0 \times \text{TV Loss} $       |    29.01     |    0.934     |    27.10     |    0.891     |    25.08     |    0.854     |
> | $ \text{MSE} + 10^{-1} \times \text{TV Loss} $   |    27.48     |    0.946     |    26.17     |    0.902     |    26.13     |    0.909     |
> | $ \text{MSE} + 10^{-2} \times \text{TV Loss} $   |    34.71     |    0.977     |    32.79     |    0.973     |    28.99     |  **0.940**   |
> | $ \text{MSE} + 10^{-3} \times \text{TV Loss} $   |    30.11     |    0.948     |    28.24     |    0.921     |    26.63     |    0.895     |
> | $ \text{MSE} + 10^{-4} \times \text{TV Loss} $   |    28.78     |    0.929     |    26.88     |    0.880     |    26.06     |    0.892     |
> | **Ours**              |  **43.61**   |  **0.996**   |  **40.07**   |  **0.977**   |  **34.72**   |    0.888     |
>
>
> **The above numerical analysis illustrates our method is still better than the comparison method (with TV loss) even without applying TV loss. Our analysis is as follows:**
>
> 1. We believe the underlying reason is that the objects in CaCO$_ 3$ exhibit varying rotational orientations. As a result, none of the edges are axis-aligned (i.e., neither parallel to the x-axis nor the y-axis), and the TV loss struggles to handle such slanted edges. Because we designed an automatic rotation function for our NAB, this situation can be handled very well.
>
> 2. More generally, **our method can be extended to non-rectangular cases and can adapt to various curve structures, which is a capability that TV lacks.** The conclusion can supported by the discussions in classic literature[1]: “*TV-minimizing techniques tend to smooth out rough or non-radially symmetric boundaries, and can consequently result in the deformation of boundaries in the function*” and literature[2]:"*Thus, especially corners in objects, but also regions with strongly varying curvature are resolved badly*". Because the coding part of our work will form corners naturally, so they can be handled very well by our method.
>
> [1] Strong D, Chan T. Edge-preserving and scale-dependent properties of total variation regularization[J]. Inverse problems, 2003, 19(6): S165.
>
> [2] Grasmair M, Lenzen F. Anisotropic total variation filtering[J]. Applied Mathematics & Optimization, 2010, 62(3): 323-339.

---

> ### Author Response · Authors · 2025-11-24
> **Response to Reviewer 4f2T (Questions 2)**
>
> >**Question 2:**
> > “The results in Table 2 appear unusual, as the INR method shows better performance with 14 views than with 16 views. It seems counterintuitive and requires explanation.…”
>
> >**Response:**
>
> We also observed this seemingly anomalous phenomenon during our experiments. **We found that it is caused by the fact that all samples in the Workpieces dataset share the same orientation.**
>
> Specifically, with 14 views, the angular interval between scan directions is $\ \frac{\pi}{14}$ (i.e., $0, \ \frac{\pi}{14}, \ \frac{2\pi}{14}, \ \frac{3\pi}{14}, \ \dots$), whereas with 16 views, the interval is $\ \frac{\pi}{16}$ (i.e., $0, \ \frac{\pi}{16}, \ \frac{2\pi}{16}, \ \frac{3\pi}{16}, \ \dots$). As a result, the majority of scan directions do not overlap between the 14-view and 16-view settings.
>
> Objects in the Workpieces dataset contain many rectangular structures, and such structures are sensitive to scan direction. For example, when using only 2 views, if the first angle is at direction 0 and parallel to the long side of the rectangle, the second view must be as close as possible to the orthogonal direction ($\ \frac{\pi}{2}$ ) to maximize reconstruction accuracy.
>
> **Since all samples in the Workpieces dataset share the same orientation** (as shown in Fig. 6 of the main paper), most rectangular structures also share the same orientation. Consequently, every sample in the dataset exhibits the same sensitivity to viewing angles, and the 16-view setup happens to partially cover some of these “sensitive” angles.  In contrast, this issue does not appear in the CaCO₃ dataset because the samples in CaCO₃ do not share a fixed orientation.
>
> We also believe that this phenomenon is method-dependent. For example, the FBP method in Table 2 also shows abnormal performance at 14 views and 12 views. In fact, we have observed this “anomaly” in $INR_f$ and $INR_{l1}$, and additional experiments of Instant-NGP[1, 2] also exhibits the same behavior.
>
> **To further validate this hypothesis, we rotated every sample in the Workpieces dataset by $\ \frac{\pi}{12}$  in the same direction**, producing a new dataset we call Workpieces-rot-π/12. On this original dataset and rotated dataset, the performance of $INR_f$ is:
>
> Table 2. Reconstruction results of $INR_f$ on the Workpieces dataset and Workpieces-rot-π/12 dataset.
> | Dataset              | 16-view PSNR | 16-view SSIM | 14-view PSNR | 14-view SSIM | 12-view PSNR | 12-view SSIM |
> |----------------------|--------------|--------------|--------------|--------------|--------------|--------------|
> | Workpieces           | 33.34        | 0.911        | 34.03        | 0.912        | 27.37        | 0.787        |
> | Workpieces-rot-π/12  | 30.43        | 0.861        | 30.16        | 0.829        | 26.53        | 0.743        |
>
> The performance of Instant-NGP also  exhibits the same pattern:
>
> Table 3. Reconstruction results of Instant-NGP on the Workpieces dataset and Workpieces-rot-π/12 dataset.
> | Dataset     | 16-view PSNR | 16-view SSIM | 14-view PSNR | 14-view SSIM | 12-view PSNR | 12-view SSIM |
> |------------------------|--------------|--------------|--------------|--------------|--------------|--------------|
> | Workpieces             | 31.17        | 0.904        | 31.95        | 0.914        | 28.38        | 0.832        |
> | Workpieces-rot-π/12    | 30.65        | 0.900        | 30.48        | 0.901        | 28.83        | 0.886        |
>
> **Table 2 and Table 3 shows that the "anomaly" disappeared after rotating the Workpieces dataset. Our method NAB does not exhibit this anomalous behavior. Because NAB includes an automatic rotation mechanism, and the underlying representation is bin-based.**
>
> [1] Müller, Thomas, et al. "Instant neural graphics primitives with a multiresolution hash encoding." ACM transactions on graphics (TOG) 41.4 (2022): 1-15.
>
> [2] Zha R, Zhang Y, Li H. NAF: neural attenuation fields for sparse-view CBCT reconstruction[C]//International Conference on Medical Image Computing and Computer-Assisted Intervention. Cham: Springer Nature Switzerland, 2022: 442-452.

---

> ### Author Response · Authors · 2025-11-24
> **Response to Reviewer 4f2T (Questions 3)**
>
> >**Question 3:**
> > “The reconstruction results of INR in Figure 6 appear excessively smooth. Could this indicate potential overfitting to the training data?.…”
>
> >**Response:**
>
> We believe that the smooth ("blurrines" would be a more accurate description) produced by $INR_f$ is caused by its insufficient generation of “harmonic compositions” to represent the image, as described in Section 3 “MOTIVATION” of the original paper.
>
> After zooming in on the reconstructed image of $INR_f$ in Figure 6 of the original paper, we can also observe that $\mathrm{INR}_f$ produces artifacts in the background of the objects, which further reflects its limited representational capability.
>
> This can be explained intuitively. During training, $INR_f$ continuously generates various smoothing functions, which lie on a manifold $\mathcal{A}$. The final training result corresponds to the smoothing function on $\mathcal{A}$ that is closest to the real industrial object (including rectangular shapes). Consequently, the reconstructed image will inevitably contain some blurring or artifacts.

---

> ### Author Response · Authors · 2025-11-24
> **Response to Reviewer 4f2T (Questions 4)**
>
> >**Question 4:**
> > “The evaluation on medical datasets should be conducted to validate the robustness…”
>
> >**Response:**
>
> Section A.2 “Limitations and Future Work” of our original paper also notes that this work was not evaluated on medical data, as it was primarily motivated by the needs of industrial CT. **Nevertheless, to further demonstrate the robustness of our method, we have added Section A.7 in the Appendix of the revised paper, where we provide both visualization and numerical results for medical reconstruction.**  Please refer to the revised paper in the OpenReview system.
>
> Specifically, we constructed a Medical-Axial dataset comprising 40 different slices, which are derived from the axial plane of the first four lung volumes in the training set of the Medical Segmentation Decathlon[1]. Specifically, the four lung volumes are labeled Lung\_001, Lung\_003, Lung\_004, and Lung\_005 in the raw data. For each volume, we extracted 10 slices at 5-slice intervals starting from the 5th slice (i.e., the slices indexed 5, 10, 15, ... 50). The projection of the Medical-Axial dataset is obtained from parallel-beam geometry with sparse view under 48 different angles.
>
> To compare performance, we use INR based on RFC as baseline. For fair comparison, we ensure that the depth, width, activation function of the FCN, the length of the encoded output vector of the encoder, total number of training epochs, and the optimization method are exactly the same as those in our method.  Because medical data contains more detailed textures than industrial data, we use a ten-layer FCN for both the comparative method and our approach.
> For simplicity, we abbreviate the comparative method as $INR_ {RFC}$.
>
> Table 4: Reconstruction results on the whole Medical-Axial dataset and different sub-datasets.
>
> | Methods        | Lung_001 PSNR | Lung_001 SSIM | Lung_003 PSNR | Lung_003 SSIM | Lung_004 PSNR | Lung_004 SSIM | Lung_005 PSNR | Lung_005 SSIM | Whole dataset PSNR | Whole dataset SSIM |
> |----------------|---------------|---------------|---------------|---------------|----------------|----------------|-----------------|----------------|---------------------|---------------------|
> | $INR_{RFC}$    | 31.82         | 0.890         | 34.03         | 0.882         | 33.71          | 0.824          | 31.34           | 0.877          | 32.72              | 0.868              |
> | **Ours**       | **33.00**     | **0.909**     | **34.49**     | **0.893**     | **34.25**       | **0.834**       | **31.39**       | **0.886**       | **33.28**          | **0.881**          |
>
>
> **Quantitative evaluation** Table 4 reports the average PSNR/SSIM on the different sub-datasets (10 slices from sub-dataset Lung\_001, 10 slices from sub-dataset Lung\_003, 10 slices from sub-dataset Lung\_004, and 10 slices from sub-dataset Lung\_005) and whole Medical-Axial dataset. We find that **replacing random Fourier encoding with our NAB boosts $INR_ {RFC}$ by 0.56 dB in whole Medical-Axial dataset**.
>
> **Qualitative evaluation** The qualitative evaluation of reconstruction is shown in Figure 9 of the revised paper. We found that our method achieves good performance even in medical data with complex shapes that do not contain rectangular objects, and our results do not show wave-like artifacts.
>
> **The experiments on the aforementioned medical dataset demonstrates that our method is robust on medical dataset.**
>
> [1] Antonelli M, Reinke A, Bakas S, et al. The medical segmentation decathlon[J]. Nature communications, 2022, 13(1): 4128.

---

> ### Author Response · Authors · 2025-11-26
>
> I hope the above response has addressed the reviewer’s current concerns. If anything is unclear or if there are any mistakes in our description, please feel free to correct us, and we will do our best to make the necessary revisions. We also hope that the reviewer will feel free to contact us  if there are any additional concerns.

---

### Official Review · Reviewer_VTfR · 2025-11-01

**Soundness:** 3
**Presentation:** 3
**Contribution:** 3
**Rating:** 6
**Confidence:** 3

**Summary:**

The paper proposes a new encoding method used in implicit neural representation, which is learnable and can be used for image with simple structure. The experiments show the proposed method is effectively in sparse view CT reconstruction.

**Strengths:**

The self-supervised method can reconstruct the CT image with high quality.
Mathematical limits of the encoding result was given.

**Weaknesses:**

The proposed method has not evaluated with complex image. Such as medical image or industrial CT image with complex structure.

If the method can be adapted to 3D image reconstruction.

For the propose method needs about 30000 iterations. For the industrial scene，the time costs of proposed method is significantly. The comparison of time costs and iteration number with INR (Random Fourier Features) should be given.

Comparison with some other method is necessary, such as data-dirven tight frame method、DIP+TV method, also diffusion based SOTA method, DPS.

**Questions:**

What’s the performance of the proposed method with full view or less view (such as 8).

How to select the hyper parameter M.

---

> ### Author Response · Authors · 2025-11-25
> **Response to Reviewer VTfR (Weaknesses 1)**
>
> **We appreciate Reviewer VTfR’s for the constructive suggestions and questions. Their comments provide important guidance that will help us refine our work. Our response to the comments are as follows:**
>
> >**Weaknesses 1:**
> > “The proposed method has not evaluated with complex image. Such as medical image or industrial CT image with complex structure…”
>
> >**Response:**
>
> We have added Section A.7 in the Appendix of the revised paper, where we provide both visualization and numerical results for medical reconstruction.**  Please refer to the revised paper in the OpenReview system.
>
> For convenience, we have placed the text and numerical results here (the paper includes image visualizations):
>
> Specifically, we constructed a Medical-Axial dataset comprising 40 different slices, which are derived from the axial plane of the first four lung volumes in the training set of the Medical Segmentation Decathlon[1]. Specifically, the four lung volumes are labeled Lung\_001, Lung\_003, Lung\_004, and Lung\_005 in the raw data. For each volume, we extracted 10 slices at 5-slice intervals starting from the 5th slice (i.e., the slices indexed 5, 10, 15, ... 50). The projection of the Medical-Axial dataset is obtained from parallel-beam geometry with sparse view under 48 different angles.
>
> To compare performance, we use INR based on RFC as baseline. For fair comparison, we ensure that the depth, width, activation function of the FCN, the length of the encoded output vector of the encoder, total number of training epochs, and the optimization method are exactly the same as those in our method.  Because medical data contains more detailed textures than industrial data, we use a ten-layer FCN for both the comparative method and our approach.
> For simplicity, we abbreviate the comparative method as $INR_ {RFC}$.
>
> Table 4: Reconstruction results on the whole Medical-Axial dataset and different sub-datasets.
>
> | Methods        | Lung_001 PSNR | Lung_001 SSIM | Lung_003 PSNR | Lung_003 SSIM | Lung_004 PSNR | Lung_004 SSIM | Lung_005 PSNR | Lung_005 SSIM | Whole dataset PSNR | Whole dataset SSIM |
> |----------------|---------------|---------------|---------------|---------------|----------------|----------------|-----------------|----------------|---------------------|---------------------|
> | $INR_{RFC}$    | 31.82         | 0.890         | 34.03         | 0.882         | 33.71          | 0.824          | 31.34           | 0.877          | 32.72              | 0.868              |
> | **Ours**       | **33.00**     | **0.909**     | **34.49**     | **0.893**     | **34.25**       | **0.834**       | **31.39**       | **0.886**       | **33.28**          | **0.881**          |
>
>
> **Quantitative evaluation** Table 4 reports the average PSNR/SSIM on the different sub-datasets (10 slices from sub-dataset Lung\_001, 10 slices from sub-dataset Lung\_003, 10 slices from sub-dataset Lung\_004, and 10 slices from sub-dataset Lung\_005) and whole Medical-Axial dataset. We find that **replacing random Fourier encoding with our NAB boosts $INR_ {RFC}$ by 0.56 dB in whole Medical-Axial dataset**.
>
> **Qualitative evaluation** The qualitative evaluation of reconstruction is shown in Figure 9 of the revised paper. We found that our method achieves good performance even in medical data with complex shapes that do not contain rectangular objects, and our results do not show wave-like artifacts.
>
> **With the support of this dataset, validation of complex images was completed.**
>
> [1] Antonelli M, Reinke A, Bakas S, et al. The medical segmentation decathlon[J]. Nature communications, 2022, 13(1): 4128.

---

> ### Author Response · Authors · 2025-11-25
> **Response to Reviewer VTfR (Weaknesses 2)**
>
> >**Weaknesses 2:**
> > “If the method can be adapted to 3D image reconstruction…”
>
> >**Response:**
>
> Yes! Our method can be extended to 3D object reconstruction. The specific method is as follows: For any coordinate $c =(x_ c, y_ c, z_ c)$, we firstly construct the following $i^{th}$  basis functions $\hat{g}(c)_ i$:
>
> $$
>  \hat{g}(c)_ i: =\prod_{\phi \in \lbrace (x_c, y_c), (y_c, z_c), (z_c, x_c) \rbrace } (\hat{\gamma}(\phi)_ {left_ i} - \hat{\gamma}(\phi)_ {right_ i})
> $$
>
> The specific form of function $\hat{\gamma}(\cdot)$ can be found in the original formula (12) in the paper. It is worth noting that for different two-dimensional coordinate planes ($X$-$Y$ plane, $Y$-$Z$ plane, $Z$-$X$ plane), $\hat{\gamma}(\cdot)$ have the same form, but the internal parameters of the different  $\hat{\gamma}(\cdot)$  are not shared during optimization.
>
> Then, the output of the entire encoded part is $f_ E(c)$:
> $$
> f_ E(c) := [\lambda_1 \hat{g}(c)_1,\; \lambda_2 \hat{g}(c)_2,\; \dots,\; \lambda_M \hat{g}(c)_M ]^{\top}
> $$
>
> In the 3D form, the basis functions of this work will become a solid cuboid, which can be transformed into a general hexahedron by performing differentiable rotations on each face, and can also degenerate into an ellipsoid by changing the steepness of each face.

---

> ### Author Response · Authors · 2025-11-26
> **Response to Reviewer VTfR (Weaknesses 3, Weaknesses 4)**
>
> >**Weaknesses 3:**
> > “For the propose method needs about 30000 iterations. For the industrial scene，the time costs of proposed method is significantly. The comparison of time costs and.…”
>
> >**Response:**
>
> For fair comparison, both our method and the comparison methods were trained for 29,990 epochs. Furthermore, we found that the main comparison methods (e.g. INR based on Random Fourier Features) are peaked around 29,990 epochs. The peak phenomenon of INR based on Random Fourier Features can be observed in the Figure 7 in the paper. In addition, We also report the performance of our method at 19,990 epochs in Tables 1 and Table 2 of the paper.
>
> >**Weaknesses 4:**
> > “Comparison with some other method is necessary, such as data-dirven tight frame method、DIP+TV method, also diffusion based SOTA method, DPS.…”
>
> >**Response:**
>
> We appreciate the reviewer’s suggestion, and we agree that incorporating a comparative method can indeed strengthen the persuasiveness of our approach. We add DIP+TV method as below(The coefficient before TV term has been determined as 0.01 through hyperparameter search):
>
> Table 1.Reconstruction results on the CaCO₃ dataset.
>
> | Methods | 16-view PSNR | 16-view SSIM | 14-view PSNR | 14-view SSIM | 12-view PSNR | 12-view SSIM | Trainable Params |
> |---------|--------------|--------------|--------------|--------------|--------------|--------------|------------------|
> | DIP-TV  | 24.05        | 0.783        | 21.98        | 0.703        | 22.40        | 0.688        | 1.90×10⁶         |
> | Ours    | **43.61**    | **0.996**    | **40.07**    | **0.977**    | **34.72**    | **0.888**    | 2.52×10⁵         |
>
> Table 2: Reconstruction results on the Workpieces dataset.
>
> | Methods | 16-view PSNR | 16-view SSIM | 14-view PSNR | 14-view SSIM | 12-view PSNR | 12-view SSIM | Trainable Params |
> |---------|--------------|--------------|--------------|--------------|--------------|--------------|------------------|
> | DIP-TV  | 33.64        | 0.918        | 29.89        | 0.777        | 28.20        | 0.786        | 1.90×10⁶         |
> | Ours    | **36.26**    | **0.938**    | **35.23**    | **0.941**    | **32.76**    | **0.909**    | 2.52×10⁵         |
>
>
> For data-driven tight frame methods, additional training data or dictionaries are typically required. In contrast, our method is fully self-supervised and does not rely on any external data. To ensure experimental fairness, our current framework focuses on zero-external data training methods.
>
> While diffusion-based methods like DPS offer excellent performance, most rely on large-scale diffusion models (typically trained on natural images or specific CT datasets). To ensure experimental fairness, our current framework focuses on zero-external data training methods. Furthermore, the diffusion models inadequately reconstruct structural information, while INR methods preserve it effectively.

---

> ### Author Response · Authors · 2025-11-27
> **Response to Reviewer VTfR (Questions 1, Questions 2)**
>
> >**Questions 1:**
> > “What’s the performance of the proposed method with full view or less view (such as 8).”
>
> >**Response:**
>
> Our work focuses on Sparse-View CT reconstruction, and its performance surpasses that of the comparison methods. Therefore, it will also achieve good performance on full-view datasets, and may even become indistinguishable from Ground Truth, as the PSNR reaches 36dB+ (16 views) on the Workpieces dataset and 43dB+ (16 views) on the CaCO3 dataset.
>
> **At the same time, we would like to note that we have already conducted sparse-view reconstruction experiments with 12, 14, and 16 views, where our approach shows significantly stronger performance than the comparison methods. Furthermore, this method also performs well on medical datasets. We hope these results can provide some support for the effectiveness of our method.**
>
> >**Questions 2:**
> > “How to select the hyper parameter M...”
>
> >**Response:**
>
> Similar to the coding length M used in INR with random Fourier features, our selection here is also empirical. Although using a larger value would yield additional performance gains, we keep our coding length M equal to the coding length in random Fourier encoding to ensure a fair comparison.
>
> Under this setting, we believe the results sufficiently demonstrate that our method already outperforms INR approaches based on random Fourier encoding($INR_ f$, $WIRE$, $SIREN$, $INR_  {l_ 1}$) and hash encoding($INSTANT$ - $NGP$), as well as self-supervised method driven by random noise such as $DIP$ and $DIP-TV$.
>
> The results of $INR_ f$, $WIRE$, $INR_  {l_ 1}$, $DIP$ are presented in th paper. The  results of $DIP$-$TV$  are provided in the above. The results of $SIREN$,  $INSTANT$ - $NGP$ are included in the "Response to Reviewer 9B2M (Weaknesses 1) ".

---

> ### Author Response · Authors · 2025-11-27
>
> I hope the above response has addressed the reviewer’s current concerns. If anything is unclear or if there are any mistakes in our description, please feel free to correct us, and we will do our best to make the necessary revisions. We also hope that the reviewer will feel free to contact us if there are any additional concerns.

---

### Meta-Review · Area_Chair_M7ui · 2026-01-02

**Summary:**

This work proposes a novel implicit neural representation (INR) scheme for industrial CT reconstruction. The reviewers recognize that the method demonstrates strong mathematical interpretability, features a novel prior design tailored for INR tasks, has a well-structured modular design, and is presented in clear and readable writing. The primary concerns raised by the reviewers were all related to insufficient experimental validation, specifically: 1) the inclusion of additional baselines, and 2) the robustness of the method on medical datasets.

After reviewing the authors' rebuttal, I believe they have provided sufficiently comprehensive experimental validation. Notably, two reviewers who initially gave a score of 4 explicitly stated that they would be willing to increase their scores if the experimental shortcomings were addressed. Combined with a positive score of 6 from another reviewer, I therefore support the acceptance of this paper.

**Reviewer Concerns:**

The primary remaining concerns across reviewers involved: 1) inclusion of additional baselines and 2) the robustness of the method on medical datasets. After reviewing the authors' rebuttal, I believe they have provided sufficiently comprehensive experimental validation.

**Reviewer Scores:**

I believe that reviewers 9B2M and J6rB will raise their scores for the paper.

---

### Decision · Program_Chairs · 2026-01-26

Accept (Poster)